# The protein phosphatase 2A holoenzyme is a key regulator of starch metabolism and bradyzoite differentiation in *Toxoplasma gondii*

Jin-Lei Wang [1] ✉, Ting-Ting Li[1], Hany M. Elsheikha [2], Qin-Li Liang[1], Zhi-Wei Zhang[1], Meng Wang[1], L. David Sibley[3] & Xing-Quan Zhu [4] ✉

Phenotypic switching between tachyzoite and bradyzoite is the fundamental mechanism underpinning the pathogenicity and adaptability of the protozoan parasite *Toxoplasma gondii*. Although accumulation of cytoplasmic starch granules is a hallmark of the quiescent bradyzoite stage, the regulatory factors and mechanisms contributing to amylopectin storage in bradyzoites are incompletely known. Here, we show that *T. gondii* protein phosphatase 2A (PP2A) holoenzyme is composed of a catalytic subunit PP2A-C, a scaffold subunit PP2A-A and a regulatory subunit PP2A-B. Disruption of any of these subunits increased starch accumulation and blocked the tachyzoite-to-bradyzoite differentiation. PP2A contributes to the regulation of amylopectin metabolism via dephosphorylation of calcium-dependent protein kinase 2 at S679. Phosphoproteomics identified several putative PP2A holoenzyme substrates that are involved in bradyzoite differentiation. Our findings provide novel insight into the role of PP2A as a key regulator of starch metabolism and bradyzoite differentiation in *T. gondii*.

*Toxoplasma gondii* is a protozoan parasite that infects many warm-blooded animals and nearly a third of the world's human population[1]. This parasite has a multi-host and multi-stage life cycle, which involves sexual reproduction in the feline definitive host and asexual development in the intermediate host. *T. gondii* developmental cycle in the intermediate host involves two main distinctive stages: rapidly replicating tachyzoites and slowly dividing bradyzoites. Tachyzoites spread inside the body and differentiate into bradyzoites, which form semi-dormant tissue cysts in the neuronal and muscle cells[2]. Although tachyzoite proliferation causes pathology in the intermediate host, tachyzoite-to-bradyzoite transformation plays a key role in the parasite

pathogenesis and establishment of a long-term infection. Most acute infections are asymptomatic in immune-competent individuals, however, reactivation of a latent infection in immune-suppressed patients, such as those with HIV/AIDS or undergoing chemotherapy, can cause serious neurological illness and even death[3].

The transformation from tachyzoites to bradyzoites can be triggered in vitro by exposure to alkaline pH 8.0–8.2, high temperature (43 °C), mitochondrial inhibitors, nutrient deprivation, or immune pressure[4–8]. During differentiation, the tachyzoites-containing parasitophorous vacuole (PV) becomes modified into a cyst with a highly glycosylated wall that encloses the bradyzoites[2,9,10]. Hundreds of

[1]State Key Laboratory of Veterinary Etiological Biology, Key Laboratory of Veterinary Parasitology of Gansu Province, Lanzhou Veterinary Research Institute, Chinese Academy of Agricultural Sciences, Lanzhou, Gansu Province 730046, People's Republic of China. [2]Faculty of Medicine and Health Sciences, School of Veterinary Medicine and Science, University of Nottingham, Sutton Bonington Campus, Loughborough LE12 5RD, UK. [3]Department of Molecular Microbiology, Washington University School of Medicine in St. Louis, St. Louis, MO 63110, USA. [4]Laboratory of Parasitic Diseases, College of Veterinary Medicine, Shanxi Agricultural University, Taigu, Shanxi Province 030801, People's Republic of China. ✉e-mail: wangjinlei90@126.com; xingquanzhu1@hotmail.com

differentially expressed genes are associated with tachyzoite-bradyzoite differentiation[11–15]. A key feature of bradyzoites is their ability to accumulate starch granules in their cytosol[16–19]. Starch granules are hypothesized to serve as energy storage to sustain parasite viability in nutrient-limited environments and/or to support the parasite motility and invasion when they encounter favorable conditions[19]. Phosphorylation of several starch-metabolizing enzymes, such as glycogen phosphorylase (GP) by calcium-dependent protein kinase 2 (CDPK2) plays a role in amylopectin metabolism[18,19]. However, the regulatory mechanisms and roles of starch accumulation in *T. gondii* are still not fully elucidated.

Protein phosphorylation and dephosphorylation are regulated by protein kinases (PKs) and phosphatases (PPs), respectively, and contribute to many cellular processes including parasite invasion, motility, growth, replication, and stage-conversion[20–23]. Phosphorylation of the eukaryotic initiation factor-α (eIF2α) is crucial for the establishment and maintenance of the bradyzoite stage. For example, inhibition of eIF2α dephosphorylation induces bradyzoite formation and inhibits bradyzoite differentiation to tachyzoite, and disruption of TgIF2K-A or TgIF2K-B affects bradyzoite differentiation[24–28]. Several proteins including AP2 DNA binding proteins (ApiAP2s), which play roles in regulating the parasite differentiation, are differentially phosphorylated between tachyzoites and bradyzoites[29–33].

Protein phosphatase 2A (PP2A), a conserved serine/threonine phosphatase, plays an essential role in various cellular functions in eukaryotic organisms. The PP2A holoenzyme comprises a heterodimeric core enzyme, consisting of a 65-kDa scaffolding subunit (PP2A-A) and a 36-kDa catalytic subunit (PP2A-C), and a number of regulatory subunits (PP2A-B). The PP2A complex activities and substrate specificity are regulated by the binding of regulatory B subunits (PP2A-B) to the PP2A core enzyme[34,35]. In *T. gondii*, kinases play critical roles in many processes, including regulation of parasite invasion, motility, replication, virulence, gene transcription, metabolism, and stage differentiation[20]. However, information about protein phosphatases and their regulatory role in *T. gondii* is limited.

In this work, we investigate the function of the heterotrimeric PP2A holoenzyme in *T. gondii*. Our results show that dephosphorylation mediated by PP2A plays important roles in the parasite growth, virulence, starch metabolism, and bradyzoite differentiation.

## Results

### PP2A holoenzyme is conserved in *T. gondii*

Homology searches of *Homo sapiens* PP2A holoenzyme in the ToxoDB database (https://toxodb.org) revealed the presence of a putative scaffolding subunit (PP2A-A, TGME49_315670) and a putative catalytic subunit (PP2A-C, TGME49_224220), and two putative regulatory subunits (TGME49_246510 known as PP2A-B and TGME49_200400 known as TgPR48 due to its homology to *Homo sapiens* PR48)[36]. To localize the subunits of PP2A, a 6HA epitope tag was fused to the C-terminus of the four subunits in a *T. gondii* type 1 RH strain and a 6Myc epitope tag into *T. gondii* type 2 Pru strain (Supplementary Fig. 1a, b). In the tachyzoite and bradyzoite stages, PP2A-C, PP2A-A and PP2A-B were detected in the cytoplasm (Fig. 1a, b). In tachyzoites at the G1 phase, PR48 was almost undetected by IFA, while in the S and M phases PR48 accumulated in the apical region of the daughter tachyzoites (Supplementary Fig. 1c).

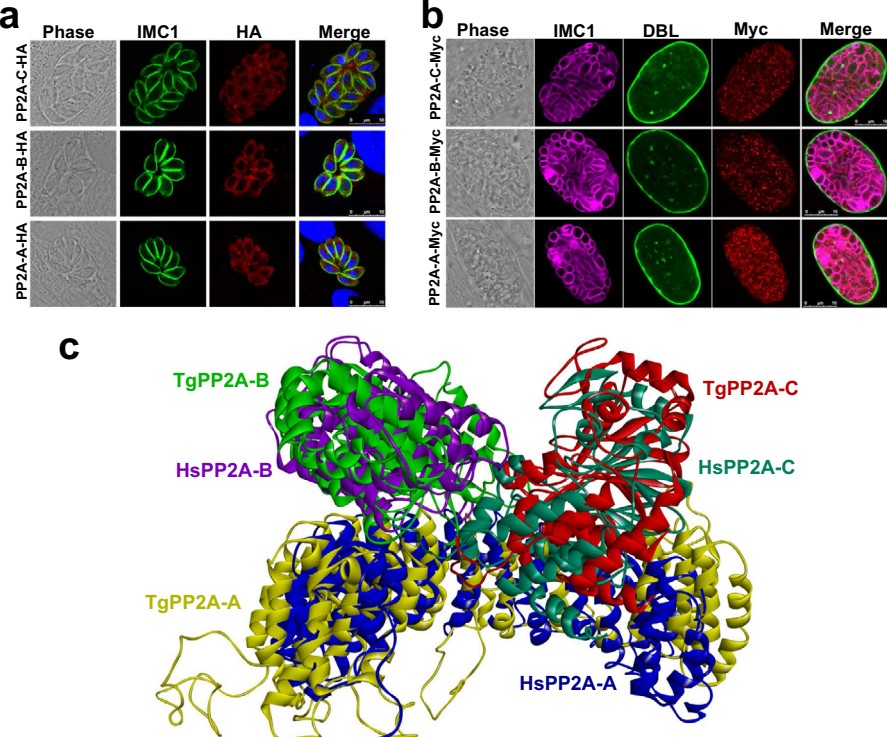

**Fig. 1 | Characterization of PP2A holoenzyme of *Toxoplasma gondii*. a** Indirect immunofluorescence of *T. gondii* RH tachyzoites expressing PP2A-C-6HA, PP2A-B-6HA and PP2A-A-6HA under the control of their endogenous promoters. Cells were fixed at 24 h post infection and stained with anti-IMC1 (green) antibody, anti-HA (red) antibody, and Hoechst DNA-specific dye (blue). Scale bar, 10 μm. **b** Indirect immunofluorescence of Pru bradyzoites expressing PP2A-C-6myc, PP2A-B-6myc and PP2A-A-6myc under the control of their endogenous promoters. Parasites were grown in HFF cells under bradyzoite-inducing alkaline conditions for 7 days and stained with FITC-*Dolichos biflorus* lectin (DBL) (green), anti-Myc (red) antibody and anti-IMC1 (magenta) antibody. Scale bar, 10 μm. **c** The overlap of *Homo sapiens* HsPP2A (PDB: 2iae, https://www.rcsb.org/structure/2IAE) structure and the predicted *T. gondii* TgPP2A holoenzyme. The protein structure homology-modeling program SWISS-MODEL was used to construct the structure of the *T. gondii* PP2A holoenzyme. *Homo sapiens* PP2A-A, PP2A-B, and PP2A-C were used as templates for TgPP2A-A, TgPP2A-B, and TgPP2A-C modeling, respectively. The three predicted PP2A subunits were constructed using ZDOCK module in the Discovery Studio.

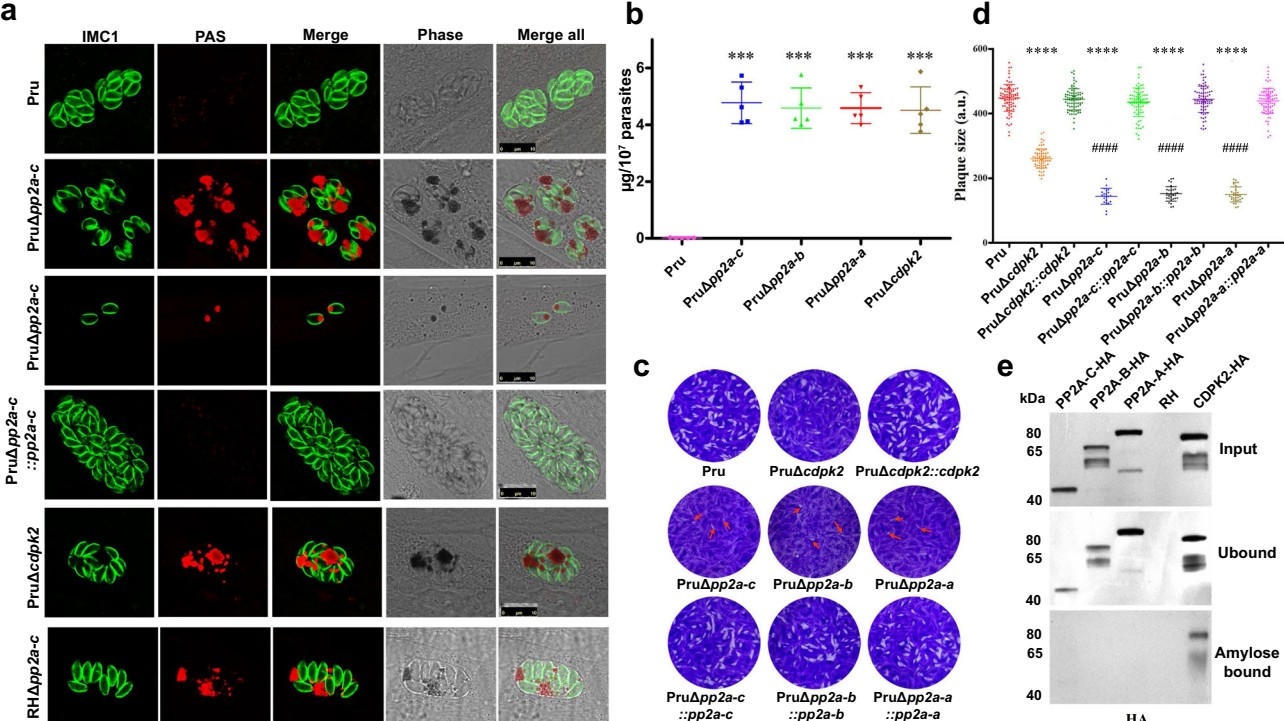

**Fig. 2 | PP2A holoenzyme is essential for starch metabolism and lytic life cycle of *Toxoplasma gondii*. a** The wild-type Pru, PruΔ*cdpk2*, Δ*pp2a-c* and the complemented strains were inoculated into HFF cells under normal culture conditions and the amylopectin was stained with PAS (red), followed by immunofluorescence detection of tachyzoites with anti-IMC1 antibody (green). The third row shows the amylopectin accumulation in the extracellular or newly invaded PruΔ*pp2a-c* tachyzoites. Scale bar, 10 μm. **b** Quantification of amylopectin levels in the indicated strains. Data represents the mean ± SD from five independent experiments. Statistical significance tested by two-tailed, unpaired *t* test with Welch's correction. ***P value = 0.0001 (Pru vs PruΔ*pp2a-c*), ***P value = 0.0001 (Pru vs PruΔ*pp2a-b*), ***P value = 0.0001 (Pru vs PruΔ*pp2a-a*), ***P value = 0.0003 (Pru vs PruΔ*cdpk2*).

**c** Representative images of the plaque assays of the indicated strains grown under normal culture conditions for 7 days. Images are representative of three independent experiments. **d** Relative size of the plaques detected in (**c**). Data represents the mean ± SD from three independent experiments. Statistical significance tested by two-tailed, unpaired *t* test with Welch's correction and the indicated strains were compared with Pru ****P value < 0.0001 or PruΔ*cdpk2* strain ####P value < 0.0001. **e** Western blotting analysis of amylose binding of PP2A-C-6HA, PP2A-B-6HA, PP2A-A-6HA, and CDPK2-3HA. Input, unbound, and amylose-bound fractions were stained with anti-HA antibody. CDPK2 was used as a positive control for amylose binding. Source data are provided as a Source data file.

To determine which regulatory subunit binds to the core enzyme, immunoprecipitation (IP) was performed. First, a 6Myc epitope tag was fused to the C-terminus of PP2A-B and PP2A-A of Pru strains that were then induced to form bradyzoites over 4 days in an alkaline culture condition. The PP2A subunits were immunoprecipitated with anti-Myc-conjugated magnetic beads and then analyzed using LC-MS. The results showed that PP2A-C, PP2A-B, and PP2A-A formed heterotrimeric complexes, while PR48 did not bind with the PP2A core enzyme (Supplementary Data 1). To confirm this interaction, a 6Myc epitope tag was fused to the C-terminus of the PP2A-C, PP2A-B, and PR48 subunits in the RHPP2A-A-6HA strain background (Supplementary Fig. 1b, d). Immunoprecipitation revealed that PP2A-C and PP2A-B were bound with PP2A-A. However, PR48 did not bind with the PP2A-A (Supplementary Fig. 1e), indicating that *T. gondii* PP2A holoenzyme consists of PP2A-A, PP2A-B, and PP2A-C subunits in both tachyzoite and bradyzoite stages. To provide the structural basis for understanding *T. gondii* PP2A function and regulation, crystal structure of a heterotrimeric PP2A holoenzyme was modeled based on the *Homo sapiens* PP2A holoenzyme[34]. The model showed a high similarity of PP2A between *T. gondii* and *Homo sapiens* (Fig. 1c).

## Absence of PP2A holoenzyme causes abnormal amylopectin accumulation

To investigate the function of PP2A holoenzyme, the gene encoding the catalytic subunit PP2A-C was disrupted by using CRISPR-Cas9 in a type 1 RH strain and a cyst-forming type 2 Pru strain (Supplementary

Fig. 2a, b). Disruption of PP2A-C led to the accumulation of granules in the intracellular tachyzoites, a feature that was more prominent in type 2 strain (Supplementary Fig. 2c). Analysis of the Δ*pp2a-c* line by transmission electron microscopy showed granular deposits corresponding to semi-crystalline polysaccharide granules (PGs), which were similar to that observed in Δ*cdpk2* mutant, suggesting that these granules are amylopectin (Supplementary Fig. 2d)[19]. Periodic Acid-Schiff (PAS) staining was performed to confirm the amylopectin nature of the granules. Δ*pp2a-c* parasites showed very pronounced staining of multiple puncta; however, the parental and complemented strains showed no staining or only very small puncta (Fig. 2a). In addition, disruption of either scaffolding subunit PP2A-A or the regulatory subunit PP2A-B caused abnormal amylopectin accumulation (Supplementary Fig. 2b, d, e). These findings are supported by quantitative analysis of amylopectin abundance in the wild-type, mutant, and complemented strains (Fig. 2b) and showed that PP2A holoenzyme is involved in amylopectin metabolism. Interestingly, a plaque assay that assesses the parasite growth over multiple lytic cycles revealed fewer and smaller plaques formation in the Δ*pp2a* strain compared with their parental, complemented strains and even with Δ*cdpk2* strain (Fig. 2c, d), suggesting that PP2A holoenzyme plays more roles in addition to regulating amylopectin metabolism.

All previously described proteins involved in amylopectin metabolism such as CDPK2, GP, pyruvate phosphate dikinase (PPDK), and alpha-glucan water dikinase (WDK) are starch-binding proteins[19]. To test whether PP2A needs to bind to amylopectin to directly regulate

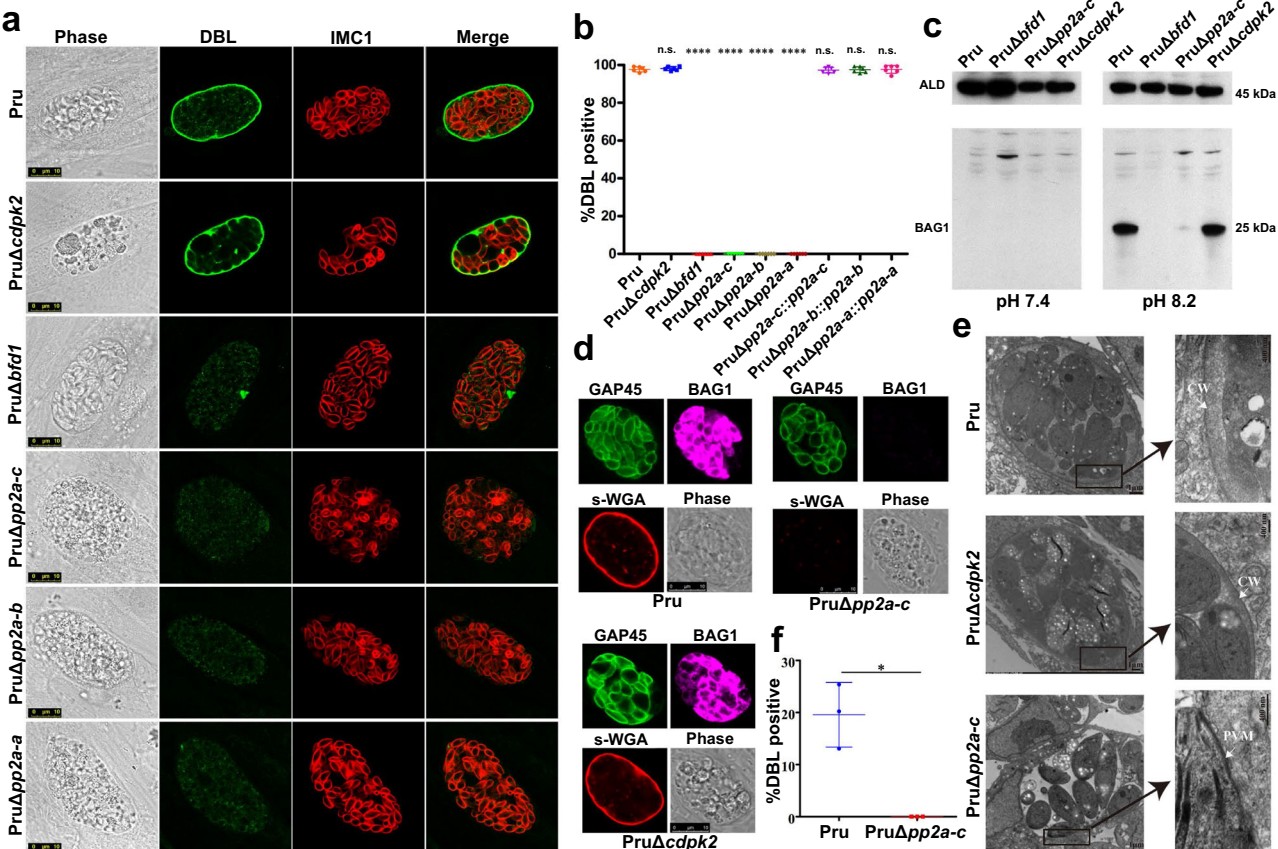

**Fig. 3 | PP2A holoenzyme is essential for bradyzoite differentiation in cell culture. a** The indicated parasite strains were allowed to infect HFF cells for 4 h followed by incubation in an alkaline culture medium without $CO_2$ for 4 days for induction of bradyzoites. Parasites were stained with anti-IMC1 antibody (red) and bradyzoite cyst wall was detected by FITC-*Dolichos biflorus* lectin (DBL) (green). PruΔ*Bfd1* was used as a negative control for stage conversion. Scale bar, 10 μm. **b** Quantification of differentiation in wild-type, knockout, and complemented strains following 4 days of exposure to alkaline stress. The mean ± SD was plotted for six biological replicates, with percentage of DBL positive vacuoles calculated based on at least 100 vacuoles per replicate. Statistical significance tested by Student's one-tailed *t* test and the indicated strains were compared with Pru strain ****P value < 0.0001, n.s. = not significant. **c** Western blotting analysis of the tachyzoites lysates (incubated for 2–3 days under normal culture conditions) and bradyzoites (incubated for 4 days under high alkaline culture conditions) by anti-BAG1

antibody. **d** The indicated parasites were added to HFF cells for 4 h to allow the parasites to invade host cells, followed by incubation in an alkaline culture medium without $CO_2$ for 4 days for induction of bradyzoites. Parasites were detected by anti-GAP45 antibody (green) and bradyzoite cysts were stained with succinylated wheat germ agglutinin (s-WGA) (red) and the bradyzoites specific anti-BAG1 antibody (magenta). Scale bar, 10 μm. **e** Transmission electron micrograph of parasites incubated for 4 days under high alkaline culture conditions. Arrowheads point to cyst wall (CW) or parasitophorous vacuole membrane (PVM) in the insert. **f** Quantification of spontaneously differentiated bradyzoites of Pru and PruΔ*pp2a-c* strains following 48 h growth in differentiated myotubes. The mean ± SD was plotted for three biological replicates, with the percentage of DBL positive vacuoles calculated based on examining at least 100 vacuoles per sample. (Student's one-tailed *t* test, *P value = 0.0159). Source data are provided as a Source data file.

starch stores, cell lysates from tachyzoites expressing PP2A-HA were analyzed using amylose-resin binding assay and the distribution of PP2A subunits in the RHΔ*cdpk2* or RHΔ*pp2a-c* tachyzoites was determined by IFA analysis. As expected, western blotting showed that CDPK2-HA bound to the amylose resin, while PP2A-HA exhibited no binding (Fig. 2e). Further IFA analysis confirmed that PP2A subunits was not associated with amylopectin granules while CDPK2 was bound with amylopectin granules in RHΔ*pp2a-c* (Supplementary Fig. 2f, g). These results suggest that PP2A may regulate starch metabolism via dephosphorylating one or more key starch-binding proteins.

**PP2A holoenzyme regulates bradyzoite differentiation**
Loss of CDPK2 causes massive amylopectin accumulation and death of bradyzoites without affecting bradyzoite differentiation[19]. We wondered whether PP2A and CDPK2 play similar roles in regulating amylopectin in the bradyzoite stage. We investigated the effect of PP2A deletion on tachyzoite-bradyzoite differentiation in response to alkaline stress by culturing parasites in an alkaline (pH 8.2) medium for 4 days. The frequency of differentiation was determined based on

positive staining of bradyzoite-specific cyst wall glycoproteins using *Dolichos biflorus* lectin (DBL)[9,10]. As anticipated, nearly all Pru and PruΔ*cdpk2* tachyzoites were successfully transformed into bradyzoites, as evident by DBL staining (Fig. 3a, b). In contrast, PruΔ*pp2a-c*, PruΔ*pp2a-b* and PruΔ*pp2a-a* were not markedly stained with DBL. Interestingly, compared with PruΔ*bfd1*, a mutant that completely fails to differentiate[37], only ~5% of Δ*pp2a* vacuoles were partly or weakly positive for DBL (Supplementary Fig. 3a). Western blotting and IFA of BAG1, a bradyzoite specific antigen[38], showed that this bradyzoite specific protein was very weakly expressed in Δ*pp2a-c* in response to alkaline stress (Fig. 3c, d). In addition, the cyst wall was also heavily stained by the chitin-binding lectin succinylated wheat germ agglutinin (s-WGA), which selectively binds to N-acetylglucosamine-decorated structures[39], in the Pru and PruΔ*cdpk2* strains; however, no staining was detected in PruΔ*pp2a-c* (Fig. 3d). Complementation of PP2A knockout restored the normal differentiation and abundance and staining of amylopectin (Supplementary Fig. 3b).

To further characterize the role of PP2A in the regulation of *T. gondii* stage differentiation and amylopectin stores, mutant strains

were differentiated in alkaline medium for 6 days. Pru∆*cdpk2* bradyzoites excessively accumulated amylopectin granules from day 2 to day 6. By contrast, ∆*pp2a* exhibited a mild increase of amylopectin granules. In addition, the size of ∆*pp2a* vacuoles was significantly larger than that of the wild-type Pru or Pru∆*cdpk2* strains grown under identical conditions. This result suggests that the mild increase in amylopectin granules in ∆*pp2a* was attributed increased parasite growth rather than tachyzoite-bradyzoite conversion (Supplementary Fig. 3c). On the other hand, the intensity of vacuoles stained positive for DBL did not increase during bradyzoite induction, and some ∆*pp2a* mutants egressed from the vacuoles. In addition, transmission electron microscopy revealed normal cyst wall ultrastructure with an amorphous granular layer underneath the cyst membrane in the wild-type Pru and Pru∆*cdpk2* strains but not the Pru∆*pp2a-c* strain after 4 days incubation in an alkaline medium (Fig. 3e). Taken together, these results suggest that the partly and weakly DBL-stained vacuoles formed by the ∆*pp2a* are not proper cysts.

In skeletal muscle cells such as murine C2C12 and human KD3 myotubes, tachyzoites can convert spontaneously to bradyzoites in the absence of exogenous stress factors[40–42]. The fate of spontaneously differentiated parasites was assessed in C2C12 myotubes, and the results showed that Pru∆*pp2a-c* strain did not form cysts in C2C12 myotubes (Fig. 3f).

Transcriptomes of Pru, Pru∆*pp2a-c*, Pru∆*cdpk2,* and Pru∆*bfd1* strains grown under normal culture conditions of tachyzoites (pH 7.4) and bradyzoite-inducing conditions (pH 8.2) demonstrated that disruption of PP2A-C affected gene expression involved in tachyzoite-to-bradyzoite conversion (Supplementary Data 2). Under bradyzoite-inducing conditions, as expected, the majority of bradyzoite markers such as BAG1, LDH2, CST1, and SAG4 were significantly upregulated in Pru and Pru∆*cdpk2* strains (Supplementary Fig. 3d, e)[14]. In contrast, the expression of the majority of bradyzoite-related genes was not significantly altered in Pru∆*pp2a-c* strain following growth in bradyzoite-inducing conditions (Supplementary Fig. 3d, e).

The function of PP2A holoenzyme was also examined by the mini auxin-inducible degron (mAID) system. PP2A proteins were C-terminally tagged with haemaglutinin epitope and fused to a mini auxin degron-haemagglutinin-tag (mAID-HA) in Pru∆*ku80*::TIR1 strain (Pru::TIR1 parental) at the endogenous locus (Supplementary Fig. 4a). PP2A-C, PP2A-B, and PP2A-A proteins were efficiently degraded upon addition of auxin (indole-3-acetic acid, IAA) as shown by immunoblotting analysis (Supplementary Fig. 4b). As expected, similar results of starch metabolism and bradyzoite differentiation were observed in the mAID strains after treating parasites with IAA (Supplementary Fig. 4c). TGME49_286210 (known as PP4C)[23], which encodes a protein structurally similar to PP2A-C (PDB: 2ie3_C) was also evaluated by the mAID system. PP4C-mAID was efficiently depleted by treating parasites with IAA as detected by immunoblotting (Supplementary Fig. 4b) and IFA analysis (Supplementary Fig. 4d). Interestingly, conditional depletion of PP4C completely blocked parasite replication but did not affect the starch metabolism or bradyzoite differentiation (Supplementary Fig. 4c), suggesting this protein is not involved in PP2A holoenzyme activity.

## PP2A holoenzyme is critical for virulence and tissue cyst formation in mice

To determine the impact of PP2A loss on the parasite virulence and chronic infection, C57BL/6 female mice were infected by i.p. injection with different doses of tachyzoites. Mortality of mice infected with Pru and Pru∆*cdpk2* lines was consistent with that reported in previous virulence studies[19,43], confirming that CDPK2 is not a major determinant of tachyzoite virulence. In sharp contrast, mice infected by Pru∆*pp2a-c*, Pru∆*pp2a-b* or Pru∆*pp2a-a* strain showed lower mortality with only one mouse from Pru∆*pp2a-a* group and one from Pru∆*pp2a-b* (injected by $2 \times 10^6$ tachyzoites) succumbing to infection (Fig. 4a–c). Symptoms of

infection were also less severe in mice infected by Pru∆*pp2a-c* strain (Supplementary Fig. 5a). These results indicate that, in addition to its role in the regulation of amylopectin metabolism, PP2A is important for optimal infection.

Brain cysts were quantified in the brain of the survived mice 30 days after infection using DBL staining. As expected, all mice infected by the Pru strain produced brain cysts. Contrarily, no cysts were detected in mice infected by Pru∆*cdpk2*, Pru∆*pp2a-c*, Pru∆*pp2a-b* or Pru∆*pp2a-a* strains (Fig. 4d). Similar results were observed in Kunming mice, which are more sensitive to *T. gondii* infection[44] (Supplementary Fig. 5b–e).

Because of the few cysts detected in Pru∆*ku80* infected mice, we assessed the effect of PP2A-C deletion on the ME49 strain, which is a low-passage and more conducive to cyst formation[37]. In this background, the coding sequence of PP2A-C was disrupted by homologous recombination, producing a ME49∆*pp2a-c* strain (Supplementary Fig. 5f). Similar to the Pru∆*pp2a-c* strain, disruption of *pp2a-c* in the ME49 strain also severely impaired the parasite growth (Fig. 4e), caused unchecked starch accumulation (Supplementary Fig. 5g) and blocked parasite differentiation (Supplementary Fig. 5h). Interestingly, PP2A-C contribution to the virulence of the ME49 strain was modest as slightly higher survival rate was observed in the infected mice (Fig. 4f, g). Brain cysts from the survived mice were counted 30 days after infection. Several hundreds of brain cysts were detected in mice infected by ME49; however, no cysts were observed in the mice infected by the ME49∆*pp2a-c* strain (Fig. 4h), although *T. gondii* DNA was detected in the brains of these infected mice by using PCR targeting the *B1* gene (Supplementary Fig. 5i). Collectively, these data suggest that parasites deficient in PP2A can reach the brain but fail to form cysts.

## Functional analysis of the conserved sequence motifs of the PP2A-C

The catalytic subunits of the phosphoprotein phosphatase (PPP) family in apicomplexan protozoa contain six consensus core motifs that are conserved in eukaryotes, including GDxHG, GDxVDRG, GNHE, HxG, RG, and H (Fig. 5a and Supplementary Fig. 6)[23,45]. To determine the functional significance of these consensus core motifs, we ectopically expressed the PP2A-C (wild-type or mutant versions fused to a 3Myc tag for detection) in the Pru∆*pp2a-c* background. As expected, Pru∆*pp2a-c* tachyzoites complemented with the wild-type PP2A-C successfully restored the regulation of amylopectin metabolism and differentiation into bradyzoites (Fig. 5b). By contrast, complementation with the six PP2A-C mutants did not restore the regulation of amylopectin metabolism or bradyzoite formation. In addition, both wild-type and mutant versions of PP2A-C were not detected in the amylopectin accumulation sites (Fig. 5b). These results indicate that the consensus core motifs of PP2A-C are required for the regulation of amylopectin metabolism and bradyzoite differentiation, and demonstrate that PP2A-C is not directly associated with amylopectin.

## Comparative phosphoproteomics identifies PP2A-C-regulated processes

To identify potential substrates of the PP2A holoenzyme, comparative phosphoproteomics analysis was performed on wild-type Pru, Pru∆*cdpk2* and Pru∆*pp2a-c* strains using stable isotope labeling with amino acids in cell culture (SILAC) quantitative mass spectrometry-based proteomics, as previously described[46,47]. Briefly, Pru, Pru∆*cdpk2*, and Pru∆*pp2a-c* tachyzoites were grown in "heavy" (H), "medium" (M), and "light" (L) SILAC labeling media, respectively. After 4 days induction of bradyzoites (to obtain comparable numbers of parasites per vacuole), parasites were purified for quantitative mass-spectrometry (Fig. 6a). Differential proteins and phosphopeptides abundances of Pru versus Pru∆*pp2a-c*, and Pru∆*cdpk2* versus Pru∆*pp2a-c* were determined.

Proteomic analysis identified 584 differentially regulated proteins between Pru and Pru∆*pp2a-c*, of which 569 proteins were upregulated

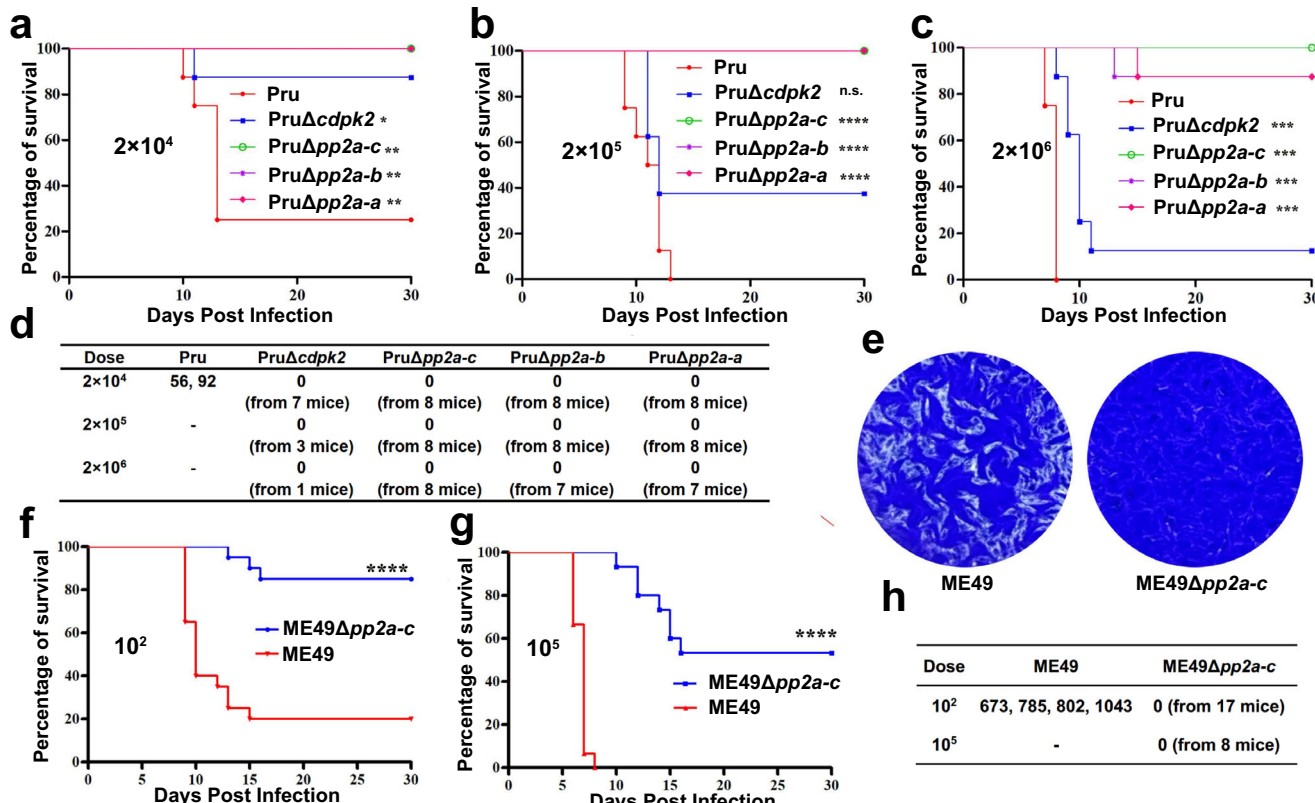

**Fig. 4 | PP2A holoenzyme is necessary for formation of brain cysts in mice.**
**a**–**c** C57BL/6 female mice were infected intraperitoneally with the designated doses of the indicated strains ($n = 8$ mice / strain), and the survival of mice was monitored for 30 days. Statistical significance tested by log rank Mantel-Cox test. Compared with Pru strain, for (**a**) group, *$P$ value = 0.024 (Pru vs Pru$\Delta cdpk2$), **$P$ value = 0.003 (Pru vs Pru$\Delta pp2a$-$c$), **$P$ value = 0.003 (Pru vs Pru$\Delta pp2a$-$b$), **$P$ value = 0.003 (Pru vs Pru$\Delta pp2a$-$a$); For (**b**) group, n.s. = not significant, ****$P$ value < 0.0001; for (**c**) group, ***$P$ value = 0.0007 (Pru vs Pru$\Delta cdpk2$), ***$P$ value = 0.0002 (Pru vs Pru$\Delta pp2a$-$c$), ***$P$ value = 0.0002 (Pru vs Pru$\Delta pp2a$-$b$), ***$P$ value = 0.0002 (Pru vs Pru$\Delta pp2a$-$a$).
**d** Number of Pru cysts in the brain tissue of the survived mice. At 30 days' post-infection, tissue cyst burden in the brain of the survived mice form (**a**–**c**) was determined by FITC-*Dolichos biflorus* lectin (DBL) staining. Cyst burden was

estimated based on the number of cysts detected in one cerebral hemisphere per mouse brain. **e** Representative images of the plaque assays of the ME49 and ME49$\Delta pp2a$-$c$ strains grown under normal culture conditions for 8 days. Images are representative of three independent experiments. **f**, **g** C57BL/6 female mice were infected intraperitoneally with the designated doses of the ME49 and ME49$\Delta pp2a$-$c$ strains, and the survival of mice was monitored for 30 days. Statistical significance tested by log rank Mantel-Cox test. ****$P$ value < 0.0001. ($n = 20$ mice / strain for the $10^2$ dose; $n = 15$ mice / strain for the $10^5$ dose). **h** The number of ME49 cysts in the brain tissue of the survived mice. At 30 days' post-infection, tissue cyst burden in the brain of the survived mice form (**f**, **g**) was determined by DBL staining. Cyst burden was estimated based on the number of cysts detected in one cerebral hemisphere per mouse brain. Source data are provided as a Source data file.

in Pru and 15 were upregulated in Pru$\Delta pp2a$-$c$. We also identified 918 differentially regulated proteins between Pru$\Delta cdpk2$ and Pru$\Delta pp2a$-$c$, of which 910 proteins were upregulated in Pru$\Delta cdpk2$ and 8 were upregulated in Pru$\Delta pp2a$-$c$. As expected, bradyzoite markers such as BAG1, ENO1, LDH2 were more abundant in Pru and Pru$\Delta cdpk2$ than in Pru$\Delta pp2a$-$c$, which is consistent with the transcriptomic data (Fig. 6b, c and Supplementary Data 3).

In the phosphopeptide-enriched samples, 10,470 phosphosites (mainly phosphoserines) in 2822 proteins were identified. For 9048 of the phosphorylation sites, we obtained quantitative proteomic data (2582 proteins) that was used to normalize the phosphorylation site changes. This approach enabled assessment of the genuine differential phosphorylation rather than changes originating from differential protein abundance between conditions (Supplementary Data 4). After normalization, we identified 670 phosphopeptides corresponding to 399 phosphoproteins with a ≥2.0 or ≤−2.0 -fold difference in the abundance between Pru and Pru$\Delta pp2a$-$c$, and 1076 differential phosphopeptides on 485 proteins between Pru$\Delta cdpk2$ and Pru$\Delta pp2a$-$c$. Overall, 225 proteins (351 phosphopeptides) were common between both datasets (Fig. 6d and Supplementary Data 5). According to the spatial data obtained from hyper LOPIT[48], most of the differentially phosphorylated protein hits are predicted to be localized in the nucleus (91/225), cytosol (32/225), and dense granules (23/225) (Fig. 6e).

Interestingly, 80 phosphosites on 58 proteins exhibited a reduced level of phosphorylation upon PP2A-C deletion (Supplementary Data 4). This result is likely to be a secondary affect caused by PP2A-C activating other protein phosphatases or suppressing kinases.

## PP2A holoenzyme dephosphorylates site on CDPK2 critical for starch metabolism
Phosphoproteomic data revealed that one serine phosphorylation site (S679) in CDPK2 protein was significantly detected in Pru$\Delta pp2a$-$c$ strain and was absent in the parental Pru strain, suggesting that CDPK2 may be dephosphorylated by PP2A-C (Supplementary Data 4). To confirm this observation, the phosphorylation status of CDPK2 in Pru$\Delta pp2a$-$c$, Pru$\Delta pp2a$-$b$ and Pru strains was examined by immunoblotting analysis. Endogenous tagging of CDPK2 in the Pru, Pru$\Delta pp2a$-$c$ and Pru$\Delta pp2a$-$b$ were performed by insertion of a 6Myc tag at the C terminus. Results of Phos-tag™ SDS-PAGE, which involves the use of a Phos-tag biomolecule that specifically binds phosphorylated proteins and hinders their migration during gel electrophoresis[49], showed that migration of CDPK2 in Pru$\Delta pp2a$-$c$ and Pru$\Delta pp2a$-$b$ was slower compared with that in Pru strain (Supplementary Fig. 7a).

To test whether only the phosphorylation status of CDPK2 influences starch metabolism, phosphomimetic (S679E) and phosphoablative (S679A) mutations were introduced into constructs encoding

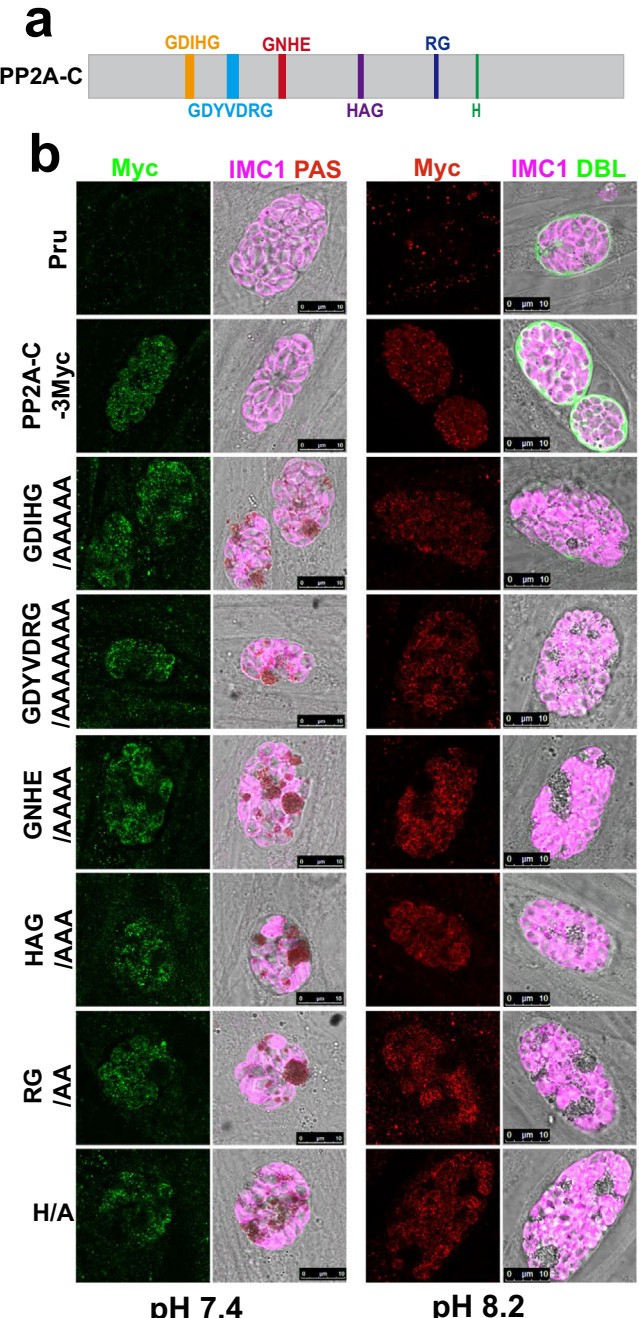

**Fig. 5 | Functional analysis of the consensus sequences of the core catalytic motif of PP2A-C. a** Schematic representation of the gene model for PP2A-C with the six consensus sequences of the core catalytic motif. **b** PAS staining of PruΔ*pp2a-c* tachyzoites expressing the wild-type or mutant version of PP2A-C fused with 3Myc tags under control of endogenous promoters, under normal culture (pH 7.4) or alkaline culture conditions (pH 8.2). The amylopectin was stained with PAS and the tagged protein was detected using anti-Myc antibody (green or red), followed by immunofluorescence detection of the parasites using anti-IMC1 antibody (magenta). Scale bar, 10 μm.

epitope-tagged CDPK2 and then transfected into PruΔ*cdpk2* strain. The expression of wild-type CDPK2-HA successfully complemented PruΔ*cdpk2* phenotype, where the parasites did not produce obvious amylopectin granules (Fig. 7a). However, phosphomimetic (S679E) and phosphoablative CDPK2 (S679A) did not complement the PruΔ*cdpk2* phenotype, with parasites accumulating large amylopectin granules. Three previously described phosphorylation sites

(S205, S701, and T706) were also investigated[50]. Phosphomimetic or phosphoablative mutation of these three phosphorylation sites did not affect starch metabolism (Supplementary Fig. 7b). Likewise, the introduction of the phosphomimetic (S679E) CDPK2 into the parental Pru strain did not alter amylopectin metabolism (Supplementary Fig. 7c). These results indicate that dephosphorylation of the phosphorylation site (S679) of CDPK2 by PP2A holoenzyme is important for the ability of CDPK2 to regulate starch metabolism.

We further examined the relation between PP2A-C and CDPK2, or GP, which is involved in the regulation of amylopectin digestion via phosphorylation of Ser25[18]. We assessed the effect of deletion of PP2A-C and CDPK2, or GP on parasite fate and amylopectin abundance. As expected, disruption of CDPK2 and GP in PruΔ*pp2a-c* strain did not significantly affect amylopectin accumulation (Supplementary Fig. 7d and Fig. 7b), suggesting that these three proteins may not operate synergistically. We also examined the effect of the phosphorylation status of CDPK2 on amylopectin accumulation in PP2A-null mutants by introducing phosphomimetic (S679E) and phosphoablative (S679A) CDPK2 in PruΔ*pp2a-c*Δ*cdpk2*. Our results showed that neither phosphomimetic nor phosphoablative mutations were able to restore amylopectin catabolism (Supplementary Fig. 7e), indicating that a balance between phosphorylation and dephosphorylation is required for maintaining CDPK2 activity.

**Identification of hyperphosphorylated proteins in PruΔ*pp2a-c* strain relevant to parasite differentiation**

Given that PruΔ*pp2a-c* tachyzoites failed to differentiate into bradyzoites, while Δ*cdpk2* mutants differentiated normally, we explored whether other substrates of the PP2A holoenzyme may be involved in bradyzoite differentiation by using quantitative phosphoproteomics. Among the most prominent substrates, transcription factors were investigated as potential regulators of differentiation. One hyperphosphorylated site (S16) in Alba1 protein was increased 334 and 301-fold in PruΔ*pp2a-c* when compared with that in Pru and PruΔ*cdpk2* strains, respectively. Interestingly, Alba1 regulates stress response and translational control of gene expression and disruption of this protein significantly diminishes the ability of the parasite differentiation[51].

The functions of the other four hyperphosphorylated protein in PruΔ*pp2a-c* strain were also evaluated (Fig. 8a). TGME49_311100 is a zinc finger (CCCH type) motif-containing protein, which contained 8 hyperphosphorylated sites in PruΔ*pp2a-c*. This protein was detected in the cytoplasm and its expression was higher in bradyzoites than in tachyzoites (Supplementary Data 3 and Supplementary Fig. 8). Disruption of TGME49_311100 did not significantly affect tachyzoite growth (Fig. 8b and Supplementary Fig. 8a), but significantly affected the ability to transform into bradyzoites (Fig. 8c, d). Unexpectedly, complementation with 3HA-tagged wild-type cDNA at the C terminus driven by the endogenous promoter did not complement the PruΔ*311100* phenotype (Supplementary Fig. 8e). However, complementation with 3HA epitope-tagged TGME49_311100 lines (CNΔ*311100*) at the N terminus restored the bradyzoite differentiation (Fig. 8c, d). In addition, we found that the endogenous TGME49_311100 gene tagged with 6HA at the C terminus phenocopied the TGME49_311100 knockouts. Taken together, these results showed that C terminal tag impaired TGME49_311100 activities.

TGME49_298610 containing a GYF domain and 4 hyperphosphorylated sites was detected in the cytoplasm (Supplementary Fig. 8b, d) and deletion of this protein inhibited tachyzoites growth (Fig. 8b) and diminished their ability to transform into bradyzoite-containing cysts (Fig. 8c, d). Due to the large size of TGME49_298610, it was difficult to amplify the coding sequence to complement the Δ*298610* strain. However, we used the mAID system to knock down the TGME49_298610 and evaluate whether depletion of TGME49_298610 could phenocopy Δ*298610* strain (Supplementary Fig. 8c, d). Results

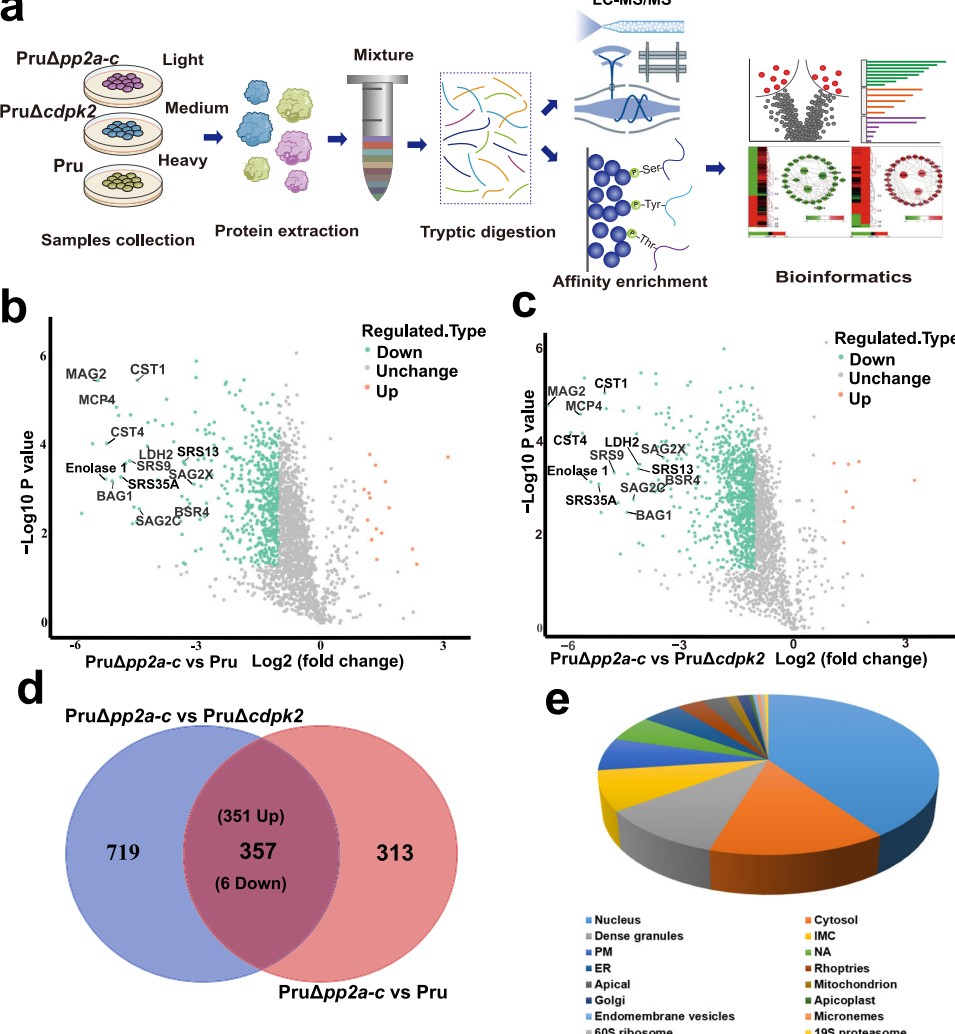

**Fig. 6 | Comparative phosphoproteomic analysis of PruΔpp2a-c, PruΔcdpk2 strains, and their parental Pru strain under alkaline culture conditions.**
**a** Schematic representation of the workflow of comparative proteomic and phosphoproteomic analyses. Parental Pru, PruΔcdpk2, and PruΔpp2a-c strains were grown in heavy, medium, or light SILAC labeling medium, respectively. After labeling, tachyzoites were allowed to differentiate to bradyzoites for 4 days under SILAC labeling medium, then the samples were harvested by needle passage, and proteins were mixed 1:1:1 before tryptic digestion. Phosphopeptides were enriched by I-MAC, and finally the samples were analyzed by LC-MS. **b**, **c** Volcano plots show significant fold changes in proteins in PruΔpp2a-c vs Pru and PruΔpp2a-c vs PruΔcdpk2 under alkaline culture conditions. Proteins with fold changes ≥2.0 or ≤−2.0 with $P < 0.05$ are highlighted in orange and laurel-green color, respectively. **d** Venn diagram of the differential phosphopeptides of PruΔpp2a-c vs Pru and PruΔpp2a-c vs PruΔcdpk2. **e** The percentage of phosphoproteins from Supplementary Data 5 relative to the total number of phosphoproteins, according to their predicted localization.

showed that mAID system-based depletion of TGME49_298610 produced a phenocopy of Δ298610 strain (Fig. 8b–d), suggesting that the lower capacity to form cysts of Δ298610 strain was due to the deletion of the TGME49_298610 rather than an off-target effect.

TGME49_224260 containing three PHD domains and TGME49_237520 containing a SANT/Myb-like DNA-binding domain were both detected in the nucleus (Fig. 8c and Supplementary Fig. 8d). TGME49_224260 and TGME49_237520 have very low fitness scores of −5.13 and −4.52, respectively, suggesting they are essential for parasite growth. To investigate their functions, mAID system was used for the conditional deletion of these two proteins. Depletion of either TGME49_224260 or TGME49_237520 by mAID system rendered the parasites unable to form plaques (Fig. 8b and Supplementary Fig. 8b–d) and significantly inhibited differentiation of tachyzoites into bradyzoite-containing cysts (Fig. 8c, d). Other transcription factors, such as ApiAP2, eukaryotic translation initiation factor, and predicted nuclear proteins, which may be involved in the transcriptional regulation during stage conversion were also hyperphosphorylated when PP2A-C was disrupted (Supplementary Data 5). These results show that

PP2A holoenzyme mediates dephosphorylation of several transcription factors that are involved in *T. gondii* differentiation.

## Discussion

The serine-threonine phosphatase, protein phosphatase 2A (PP2A), is conserved in eukaryotic organisms and regulates many cellular processes. PP2A exists in two forms: a heterodimeric core enzyme and a heterotrimeric holoenzyme. The PP2A core enzyme includes a catalytic subunit (PP2A-C) and a scaffold subunit (PP2A-A) and interacts with a regulatory subunit (PP2A-B) to form a holoenzyme[45]. The PP2A holoenzyme is conserved in *T. gondii* and involves a catalytic subunit (PP2A-C), a scaffold subunit (PP2A-A) and two putative regulatory subunits, namely PP2A-B and PR48. PP2A-C, PP2A-A, and PP2A-B form a heterotrimeric holoenzyme in *T. gondii* while PR48 does not interact with the PP2A core enzyme despite sharing some homology with human PR48. Disruption of any of the PP2A subunits caused excessive accumulation of starch and interrupted tachyzoite-bradyzoite differentiation, demonstrating the importance of heterotrimeric holoenzyme complex in *T. gondii* PP2A activity. We also

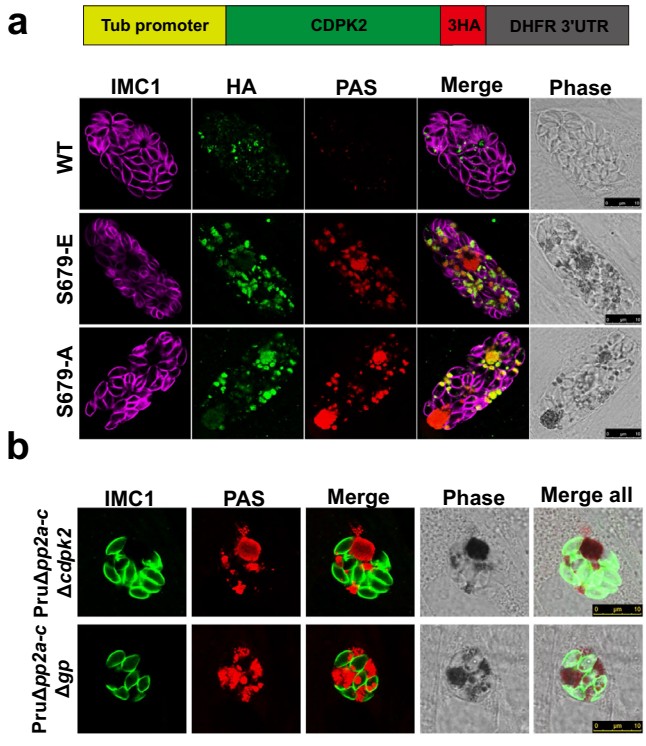

**Fig. 7 | Dephosphorylation of CDPK2 is essential for the regulation of starch metabolism. a** PAS staining of PruΔ*cdpk2* tachyzoites expressing the wild-type or phosphomimetic mutant version of CDPK2 fused with 3HA tags under control of β-tubulin promoters and normal culture conditions. The amylopectin was stained with PAS (red), the tagged protein was detected using α-HA antibody (green) and parasites were detected with anti-IMC1 antibody (magenta). Scale bar, 10 μm. **b** PAS staining of the double knockouts of PruΔ*pp2a-c*Δ*cdpk2* and PruΔ*pp2a-c*Δ*gp* strains. The amylopectin was stained with PAS (red), followed by immunofluorescence detection of parasites with anti-IMC1 antibody (green). Scale bar, 10 μm.

examined whether PP4C protein, encoded by the TGME49_286210, which is structurally similar to PP2A-C also has similar biological activity. Interestingly, the depletion of PP4C caused a growth arrest in the early phase of the lytic cycle, without any impact on starch metabolism or parasite differentiation, suggesting that PP4C is not involved in activity of the PP2A holoenzyme.

Amylopectin is found in *T. gondii* and other coccidian species, such as *Eimeria tenella* and *Cryptosporidium parvum*[52,53]. Although the accumulation of amylopectin is a hallmark of *T. gondii* bradyzoites, the function of amylopectin storage and the signaling pathways regulating its synthesis or degradation are not fully understood. In the present study, we revealed the regulatory role of PP2A holoenzyme in amylopectin metabolism in *T. gondii*. Interestingly, the level of amylopectin abundance in Δ*pp2a* tachyzoites was similar to that in Δ*cdpk2* tachyzoites. While CDPK2 binds to amylopectin[19], PP2A holoenzyme did not bind to amylopectin and was not recruited to amylopectin granules suggesting that PP2A holoenzyme may regulate starch metabolism indirectly via de-phosphorylation of enzymes in the amylopectin metabolism pathway.

The quantitative phosphoproteomic analysis identified one serine residue in CDPK2 protein (S679) whose phosphorylation was significantly detected in PruΔ*pp2a-c*, but was absent in Pru strain. Phosphomimetic (S679E) or phosphoablative (S679A) mutations were introduced into PruΔ*cdpk2* strain and the results showed that neither mutant was able to restore the amylopectin to the wild-type level. These findings suggest a mechanism by which PP2A may regulate starch metabolism via dephosphorylation of S679 of CDPK2. In addition, introduction of the S679 mutant of CDPK2 into the Pru strain did

not affect the amylopectin store, suggesting that S679 phosphorylation site does not cause a dominant-negative effect that influences the activity of wild-type CDPK2. Moreover, phosphomimetic or phosphoablative CDPK2 mutant in PruΔ*pp2a-c*Δ*cdpk2* did not alter amylopectin abundance suggesting that phosphorylation level of CDPK2 did not alter the results caused by the disruption of PP2A holoenzyme. Taken together, the balance of CDPK2 phosphorylation may be mediated by other kinases and de-phosphorylation mediated by PP2A are required for regulation of amylopectin metabolism.

CDPK2 plays a crucial role in amylopectin metabolism and deletion of CDPK2 causes excessive accumulation of amylopectin in *T. gondii* particularly in the bradyzoites[19]. We therefore explored whether the PP2A holoenzyme plays a similar role in regulating amylopectin stores in bradyzoites. Interestingly, PP2A-deficient parasites failed to differentiate in cell culture or form cysts in the brain of mice. When parasites were incubated under high pH medium for 6 days, PP2A deficient parasites exhibited a mild increase in PAS staining, while PruΔ*cdpk2* bradyzoites accumulated massive amylopectin granules. Moreover, the growth of PP2A deficient parasites was faster than that of PruΔ*cdpk2* under high alkaline conditions. In addition, in response to alkaline stress, numerous bradyzoite-specific mRNAs were not upregulated and cysts wall was not observed with PP2A disruption. These results suggest that multiple PP2A substrates are involved in the regulation of bradyzoite differentiation.

Notably, most of the phosphoproteins identified as PP2A holoenzyme-dependent and involved in the regulation of bradyzoite differentiation were nuclear proteins. The nuclear proteins containing RNA or DNA-binding domains are involved in translational control of gene expression, such as Alba1 protein, which responds to alkaline stress and its disruption significantly alters bradyzoite differentiation[51]. In our study, the phosphorylation site (S16) of Alba1 was hyperphosphorylated in PruΔ*pp2a-c*, suggesting that phosphorylation of this site might control Alba1 activity. TGME49_311100 contains three zinc finger (CCCH type) motifs that bind to DNA, RNA, protein and/or lipid substrates to mediate gene transcription, translation, mRNA trafficking, cytoskeleton organization, epithelial development, and cell adhesion[54–56]. There were 8 hyperphosphorylated sites in this protein in PruΔ*pp2a-c*, and disruption of TGME49_311100 significantly reduced parasite differentiation, suggesting that post-translational regulation of TGME49_311100 by PP2A holoenzyme contributes to parasite differentiation. Reassuringly, while this manuscript was under review, Licon et al. (bioRxiv preprint) identified TGME49_311100 (named BFD2) as an important regulator that intersects with BFD1. Loss of BFD2 reduces BFD1 protein levels and compromises the parasites' ability to fully differentiate in vitro and in mice[57]. Given that BFD2 controls BFD1 expression (only present in bradyzoites) and that PP2A is essential for BFD2 dephosphorylation, further investigation of PP2A-BFD2-BFD1 connection may illustrate the mechanisms that govern complex spatially and temporally regulated molecular events occurring during tachyzoite-to-bradyzoite differentiation.

TGME49_298610, a GYF domain-containing protein, was hyperphosphorylated in PruΔ*pp2a-c* and deletion of this protein altered parasite differentiation. The GYF domain is ~ 60-amino acid domain that plays crucial roles in protein binding[58]. The plant homeodomain (PHD) finger is a C4HC3 zinc-finger-like motif, which is found in nuclear proteins and thought to be involved in epigenetic and chromatin-mediated transcriptional regulation[59]. There are three PHD domains and one RING domain, (a specialized type of Zn-finger) in TGME49_224260, which can possibly mediate protein-protein interactions. TGME49_224260, localized in the nucleus, is essential for parasite growth and deletion of this protein reduced parasite differentiation. SANT/Myb-like DNA-binding domains are important regulators of parasite growth and differentiation, such as BFD1 which serves as a master regulator of differentiation[37,60,61]. TGME49_237520, which contains a SANT/Myb-like DNA-binding

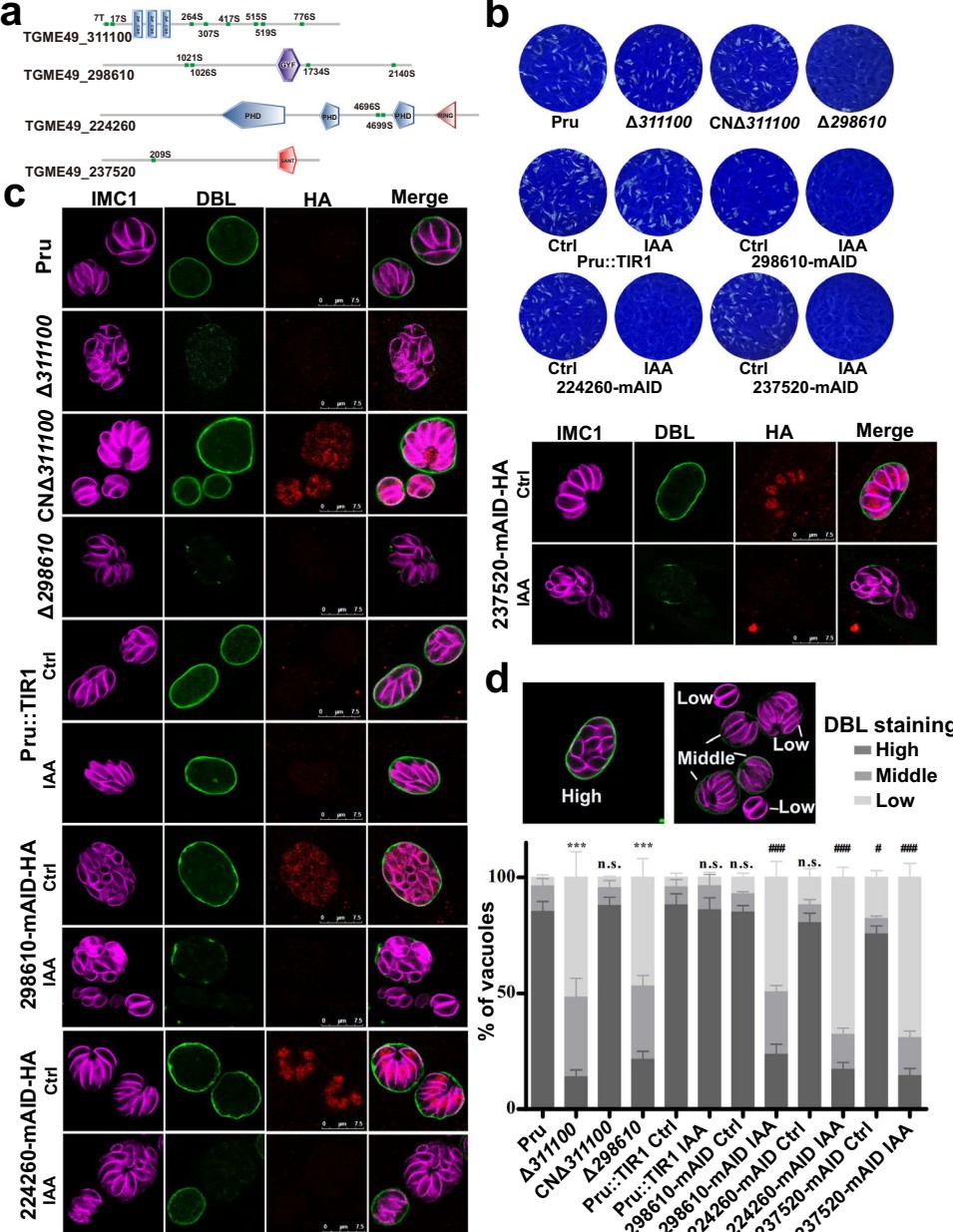

**Fig. 8 | Four hyperphosphorylated proteins in PruΔ*pp2a-c* strain are important for parasite differentiation. a** Schematic representation of the domain architecture of four hyperphosphorylated proteins in PruΔ*pp2a-c* strain.
**b** Representative images showing the results of the plaque assay of the indicated strains grown under normal culture conditions for 7 days. Images are representative of three independent experiments. **c** Representative vacuoles of differentiation after 72 h exposure to alkaline medium. FITC-labeled *Dolichos biflorus* lectin (DBL) specifically stains the differentiated vacuoles and the parasites were stained with IMC1 and mAID-HA tagged proteins were detected with anti-HA antibody. Scale bar,

7.5 μm. **d** The percentages of DBL-positive cysts were calculated based on counting at least 100 vacuoles per sample treated as described in (**c**). Data represent the means ± SD from three independent experiments. The percentage of vacuoles scoring "DBL-high" was used for comparison. Statistical significance tested by Student's one-tailed *t* test. \*\*\**P* value = 0.0002 (Pru vs Δ*311100*), \*\*\**P* value = 0.0003 (Pru vs Δ*298610*), ###*P* value = 0.0004 (Pru::TIR Ctrl vs 298610-mAID IAA), ###*P* value = 0.0002 (Pru::TIR Ctrl vs 224260-mAID IAA), #*P* value = 0.036 (Pru::TIR Ctrl vs 237520-mAID Ctrl), ###*P* value = 0.0002 (Pru::TIR Ctrl vs 237520-mAID IAA). Source data are provided as a Source data file.

domain, was important for parasite growth and differentiation and was hyperphosphorylated in PruΔ*pp2a-c*. Other DNA or RNA binding proteins, such as ApiAP2, MIF4G, RNA recognition motif-containing protein, were also hyperphosphorylated in PruΔ*pp2a-c*. These results suggest that PP2A may de-phosphorylate several proteins involved in the regulation of parasite differentiation and further investigations are required to confirm this assumption. Moreover, some proteins involved in parasite differentiation were only expressed in the bradyzoite stage, such as the BFD1 and thus were not detected in the phosphoproteomics data of PruΔ*pp2a-c* tachyzoites. Thus, further

studies are required to elucidate whether PP2A play any role in the activity of those bradyzoite-specific proteins.

The other subset of hyperphosphorylated proteins identified in PruΔ*pp2a-c* were GRA proteins, including cyst wall proteins MCP3 and CST4. During tachyzoite-to-bradyzoite transformation, the PV-resident GRAs re-localize as the tachyzoites transform to bradyzoites stage and remodel the PV into a dormant cyst[62,63]. In agreement with our study, Young et al.[64] observed widespread phosphorylation of GRAs proteins when *T. gondii* differentiates from the acute to latent stage. Whether there is a link between the hyperphosphorylated GRA

proteins and the disruption of stage conversion in Δ*pp2a* remains to be investigated.

In summary, the adverse effects caused by PP2A deletion on the parasite growth, virulence, starch metabolism and stage differentiation clearly show that PP2A-dependent dephosphorylation events play essential roles in the infectivity, pathogenesis and stage interconversion of *T. gondii* (Fig. 9). This finding adds to the growing evidence of the important role of dephosphorylation events mediated by PP2A in various cellular processes in many other organisms[45].

## Methods

### Sequence analysis and phylogenetics
Sequences related to PP2A holoenzyme and other PPP family members were analyzed as described previously[23]. Briefly, the initial hits of PPPs from *Homo sapiens* were used as query for BLAST searches against ToxoDB (https://toxodb.org) or VEupathDB (https://veupathdb.org) databases. Multiple alignments of the best hit protein sequences were generated by MUSCLE and manually edited as necessary. Gblocks 0.91b was used to select conserved blocks for construction of phylogenetic trees. Maximum likelihood phylogenies were performed by MEGA 7.0, and 100 resampling for bootstrap calculations.

### Parasite culture and transfection
*T. gondii* tachyzoites (RHΔ*ku80*Δ*hxgprt* denoted as RH or PruΔ*ku80*Δ*hxgprt* denoted as Pru) were grown in confluent monolayers of human foreskin fibroblasts (HFFs) cultured in Dulbecco Modified Eagle Medium (DMEM) supplemented with 2% fetal bovine serum (FBS), 100 mg/ml streptomycin and 10 units/ml penicillin at 37 °C and 5% $CO_2$ as previously described[65]. Prior to infection, HFFs were grown to confluency in DMEM supplemented with 10% FBS under 37 °C with 5% $CO_2$ in atmospheric air. Tachyzoites were released from heavily infected host cells by passage through a 27-gauge needle followed by filtration using a 3-μm polycarbonate membrane filter. Transgenes were performed by electroporation using an ECM 830 Square Wave electroporator (BTX) into *T. gondii* tachyzoites and stable transformants were selected by culturing in the presence of either 25 μg/ml mycophenolic acid and 50 μg/ml xanthine, 3 μM pyrimethamine, or 20 μM chloramphenicol. Single clonal lines were obtained by limiting dilution in HFF monolayers grown in 96 well tissue culture plates.

### Plasmids and primers
All plasmids were constructed by assembly of the DNA fragments by ClonExpress® MultiS One Step Cloning Kit or site-directed mutagenesis of existing plasmids using Q5® Site-Directed Mutagenesis Kit. All plasmids and primers used in this study are listed in Supplementary Data 6.

### Generation of transgenic parasite strains
**CRISPR-Cas9 mediated endogenous tagging.** A CRISPR-Cas9 plasmid was generated to establish a double stand break in the 3′-untranslated region (3′-UTR) of the gene of interest close to the stop codon as previously described[66,67]. This plasmid was co-transfected with amplicons flanked with short homology regions including a 6HA or 6Myc tag into RH or Pru strains, and the positive tagged clonal lines were confirmed by PCR and indirect immunofluorescence assay (IFA).

### CRISPR-Cas9 mediated gene knockout
Knockouts were generated by CRISPR-Cas9 mediated homologous gene replacements as described previously[68]. Briefly, a plasmid containing a drug marker DHFR or HXGPRT flanked with ~1 kb homology arms to the 5′-UTR and 3′-UTR of the respective gene, was used to amplify the homologous template. This homologous template and the

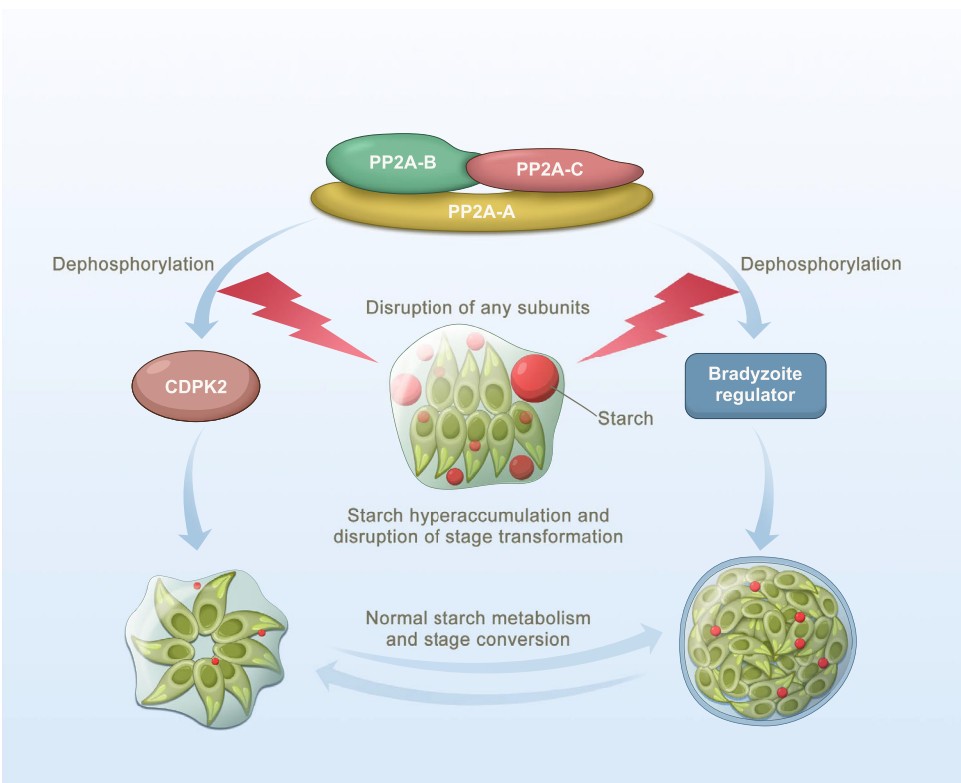

**Fig. 9 | Schematic representation of the PP2A holoenzyme as an integral component of the regulatory network mediating amylopectin metabolism and tachyzoite-to-bradyzoite transformation in *Toxoplasma gondii*.** *T. gondii* PP2A holoenzyme consists of a scaffolding subunit PP2A-A, a catalytic subunit PP2A-C and a regulatory subunit PP2A-B. The disruption of any of the PP2A subunits causes significant starch accumulation and blocks tachyzoite-bradyzoite differentiation via dephosphorylation of calcium-dependent protein kinase 2 and some bradyzoites regulators, respectively.

corresponding gene-specific CRISPR plasmid were co-transfected into purified tachyzoites, and the positive clonal lines were identified by PCR.

## CRISPR-Cas9 mediated conditional knockdown

Auxin-induced degradation system was used to conditionally knock-down essential genes as described previously[66,67]. Briefly, a CRISPR-Cas9 plasmid targeting 3′-UTR of the respective gene near the stop codon was co-transfected with amplicons flanked with short homology regions containing mAID-3HA or mAID-6HA and a DHFR marker into the Pru*Δku80*::TIR1 (referred as Pru::TIR1) strain. The insertion of mAID-3HA or mAID-6HA tag was confirmed by PCR and sequencing. For knockdown, parasites were treated with 500 μM 3-indoleacetic acid (IAA), while mock treatment included the addition of 0.1% ethanol only.

## Construction of complementation strains

To generate the PP2A mutant complementation plasmid, the individual *pp2a* promoter, coding sequences and the 3′-UTR were amplified and assembled into the pSAG1-CAT vector. Amplicons with short homologies flanking with the complementation constructs were co-transfected with CRISPR-Cas9 plasmids containing sgRNAs specifically targeting the *hxgprt* locus into the PP2A knockout strains. For complementation of Δ*cdpk2* knockouts, the promoter was changed to β-tubulin promoter as described previously[19]. For complementation of Pru*Δ311100*, an N-terminally or C-terminally 3HA-tagged cDNA copy driven by the endogenous promoter were used. The strains expressing various domain mutations were generated by Q5 mutagenesis by using the same strategy used for the construction of the complemented strain. Individual complemented clones were isolated by limiting dilution and the positive clone was determined by PCR and IFA.

## Indirect immunofluorescence analysis

HFF monolayers infected by *T. gondii* tachyzoites were fixed with 4% formaldehyde in PBS for 20 min, permeabilized with 0.2% Triton X-100 and blocked with 3% bovine serum albumin (BSA) in PBS for one hour at room temperature. The primary antibodies were added at 4 °C overnight and washed five times with PBS, followed by incubation for one hour with the secondary antibody. 4′,6-diamidino2-phenylindole (DAPI) counterstain was used to label the host cell and parasite nuclei. After washing three times with PBS, cells were imaged with a Leica confocal microscope system (TCS SP8, Leica, Germany).

## Western blotting and co-immunoprecipitation analysis

Tachyzoites were purified from the host cells and pelleted by centrifugation at $2000 \times g$ for 10 min at 4 °C. The parasite pellets were washed twice with cold PBS before treating with RIPA lysis buffer containing Protease and Phosphatase Inhibitor Cocktail. The lysates were incubated on ice for 30 min and centrifuged at $12,000 \times g$ for 10 min at 4 °C. The supernatants were prepared in Laemmli loading dye, boiled for 10 min, and separated on polyacrylamide gels by SDS-PAGE. The separated samples were transferred onto nitrocellulose membranes and blocked in 5% fat-free milk-Tris-buffered saline (TBS) supplemented with 0.2% Tween 20 (TBST) before incubation with primary and second antibodies. Phos-tag gel electrophoresis was performed according to the manufacturer's instructions (Wako Chemicals, USA). Briefly, 100 μM Phos-tag and 200 μM $MnCl_2$ were added to conventional 8% (w/v) acrylamide resolving gel and the gel was run at constant voltage. The gel was washed five times in SDS-PAGE running buffer containing 10 mM EDTA and three times in transfer buffer before transferring to a PVDF membrane for immunoblotting. Immunoprecipitations were conducted by the Pierce Magnetic Myc-Tag IP/Co-IP Kit (Thermo Fisher Scientific) according to the manufacturer's instructions[49]. Briefly, ~50 μL anti-Myc beads were added to the supernatant of the parasites lysate and incubated with end-over-end rotation at 4 °C for 3 h. The beads were collected by centrifugation, and protein was eluted from beads by the addition of 100 μL of elution buffer.

## Immunoprecipitations (IPs) for mass spectrometry

Eluted proteins were separated by SDS-PAGE until the running front had migrated ~2 cm into the gel and stained with colloidal Coomassie (InstantBlue; Expedeon). After the excision of eight horizontal gel slices per lane, proteins were in-gel digested with trypsin (Promega/Pierce) using a Janus liquid-handling system (PerkinElmer). Tryptic peptides were analyzed by liquid chromatography-mass spectrometry (LC-MS) using an Orbitrap Velos mass spectrometer coupled to an Ultimate 3000 ultrahigh-performance liquid chromatography (uHPLC) instrument equipped with an Easy-Spray nanosource (Thermo Fisher Scientific) and acquired in data-dependent mode. The RAW data were searched using the MaxQuant (v1.6.1.0) tool against the *T. gondii* ME49 databases from ToxoDB (https://toxodb.org)[69]. Peptide and protein identifications were filtered to a 1% false discovery rate (FDR), and reversed proteins, contaminants, and proteins identified by only a single modification site were removed from the data set.

## Periodic Acid-Schiff staining

Periodic Acid-Schiff (PAS) staining was performed using a modified PAS stain kit as described previously[19]. Briefly, infected cells were fixed with 4% formaldehyde, permeabilized with 0.2% Triton X-100, blocked with 3% BSA and then incubated with periodic acid solution. After washing five times with PBS, samples were treated with Schiff's reagent for 20 min at room temperature. Subsequently samples were washed with PBS and incubated with primary and secondary antibodies. The PAS-positive signal was visualized by the red fluorescence excited with a 543-nm argon laser line and images were obtained with a Leica TCS SP8 confocal microscope.

## Amylopectin quantification

The amount of amylopectin was measured in freshly egressed tachyzoites by using a modified starch colorimetric assay kit (MAK368, Sigma-Aldrich). Briefly, tachyzoites were washed five times with ice-cold PBS to remove the DMEM, and tachyzoites were suspended in ultrapure water, boiled for 10 min, cooled on ice, and centrifuged to collect the supernatants. The amounts of amylopectin in the supernatant were quantified according to the manufacturer's instructions.

## Amylopectin binding assay

Parasite lysates prepared from freshly egressed tachyzoites were used in the amylopectin binding assay[19]. Briefly, the supernatants of the parasite lysates were added to 2 mL tubes containing 100 μL of amylose beads and incubated with end-over-end rotation at 4 °C for overnight. The amylose beads were collected and the supernatants were retained as the "unbound fractions", while the beads were washed five times and re-suspended in 30 μL of SDS-PAGE sample buffer and incubated at 100 °C for 5 min, before pelleting the beads and retaining the supernatants as the "amylose bound fractions." The input, unbound and amylose-bound fractions were loaded on SDS-PAGE followed by western blotting with anti-HA antibody.

## Transmission electron microscopy

HFF monolayers infected by tachyzoites for 24–36 h or alkaline pH-induced bradyzoites for 4 days were fixed with 2.5% glutaraldehyde in 0.1 M sodium cacodylate buffer for 2 h at room temperature. Samples were washed in cacodylate buffer and post-fixed in 1% osmium tetroxide and 1.5% potassium ferricyanide for 1 h and processed as described previously[65]. TEM images were acquired with a Hitachi HT7700 electron microscope under 80 kV.

## Plaque assays

Plaque assays were performed using 12-well tissue culture plates containing HFF monolayers infected with 500 freshly egressed tachyzoites per well[65]. For conditional knockout, 500 μM IAA or 0.1% ethanol was added to the medium prior to the addition of tachyzoites. Parasites were grown for 7–9 days before staining with crystal violet. The number of plaques was counted manually and the plaque size was analyzed by ImageJ. All plaque assays were performed in triplicate.

## Bradyzoite conversion and harvest

Bradyzoite differentiation was induced by alkaline treatment of infected confluent HFF monolayers as described previously[37]. Briefly, type 2 tachyzoites were allowed to infect HFF cells for 4 h under normal culture conditions, and then the medium was replaced with alkaline RPMI-HEPES medium containing 2% FBS (pH 8.2). The cultures were incubated at 37 °C without $CO_2$ and the medium was replaced every day to maintain high alkaline pH. To harvest bradyzoites, culture medium was removed and heavily infected HFF monolayers were rinsed with PBS. Following the addition of 5–10 mL PBS, the intracellular bradyzoites were mechanically released by scraping the HFF monolayers and passing the scrapped cells through 27-gauge needles, followed by removing cell debris using a polycarbonate filter with a 5-μm pore size.

For spontaneous tachyzoite-to-bradyzoite conversion, C2C12 myoblasts (a gift from Prof. Xiaomao Luo at Shanxi Agricultural University) were cultured in DMEM supplemented with 10% FCS, 100 U/ml penicillin and 100 mg/ml streptomycin. Myoblasts were seeded in 6-well culture plates ($1 \times 10^5$ cell/well), and after 24 h growth, the medium was replaced with 2% horse serum for in vitro differentiation into mature myotubes as previously described[41]. Myoblasts were allowed to differentiate into myotubes during the over 120 h and then parasites were allowed to infect the differentiated cells for 2 days. Finally, the spontaneous cysts and differentiated myotubes were detected by staining using DBL and Myosin 4 monoclonal antibody, respectively.

## In vivo infection

Eight-week-old female Kunming or C57BL/6 mice were purchased from the Center of Laboratory Animals of Lanzhou Veterinary Research Institute. Mice were maintained under pathogen-free and controlled conditions (12/12–h dark/light cycle, 50–60% humidity, and 22 °C temperature), and had free access to sterilized food and water. For virulence studies, mice were infected intraperitoneally (i.p.) with freshly egressed tachyzoites and monitored for up to 30 days for the symptoms of infection and the humane endpoints. Plaque assays were performed to confirm that the injected tachyzoites are viable. To determine the brain cyst burden, survived mice were euthanized at 30 days after infection, and their brains were collected in PBS, homogenized for brain cyst counting by using Fluorescein labeled *Dolichos biflorus* lectin staining as described previously[37]. All procedures were approved by the Animal Research Ethics Committee of Lanzhou Veterinary Research Institute, Chinese Academy of Agricultural Sciences (LVRI-2020-13).

## RNA sequencing

Tachyzoites were used to infect HFF monolayers for 4 h and the infected monolayers were washed with DMEM, and the medium was replaced with normal medium or alkaline medium to collect tachyzoites and bradyzoites, respectively. After 48–72 h post infection, intracellular tachyzoites were harvested from the normal medium, while the intracellular bradyzoites were harvested from the alkaline medium after 4 days' infection as described above. Parasite suspensions were centrifuged for 5 min at $1000 \times g$ and the pellets were resuspended in 1 mL TRIzol to extract RNA. All extracted RNA samples were treated with RNase-Free DNase to remove residual genomic DNA

and the total RNA of each sample was qualified and quantified using a Nano Drop and Agilent 2100 bioanalyzer (Thermo Fisher Scientific, MA, USA).

For mRNA library construction, mRNA was purified by Oligo(dT)-attached magnetic beads and the purified mRNA was fragmented into small pieces with fragment buffer. Then first-strand cDNA was generated using random hexamer-primed reverse transcription, followed by a second-strand cDNA synthesis. The cDNA fragments were subjected to end repair, A-tailing, and adapter ligation and amplified by PCR. The products were purified by Ampure XP Beads and dissolved in EB solution. The product was validated on the Agilent Technologies 2100 bioanalyzer for quality control. The double-stranded PCR products from the previous step were heat denatured and circularized by the splint oligo sequence to get the final library. The single-strand circle DNA (ssCir DNA) was formatted as the final library. The final library was amplified with phi29 to make DNA nanoball (DNB), which had more than 300 copies of one molecular, DNBs were loaded into the patterned nanoarray and pair end 100 bases reads were generated on BGIseq500 platform.

The sequence raw data was filtered using SOAPnuke (v1.5.2) to remove reads containing sequencing adapters, low-quality reads and reads with unknown base ('N' base) ≥5%. The clean reads were obtained and stored in FASTQ format. The clean reads were then mapped to the reference *T. gondii* ME49 genome using hierarchical indexing for spliced alignment of transcripts (HISAT2) (v2.0.4)[70]. Reads per kilobase per million mapped reads (RPKM) method was used to calculate the relative gene expression. DESeq2 software was used to determine gene expression and identify differentially expressed genes (DEGs)[71]. The volcano plots were constructed using R based on the gene expression levels in different samples. Genes with a $\log_2$ fold change of ≥±1.0 and $P$ value of <0.05 were considered significant.

## Proteomics studies

**SILAC labeling and sample preparation.** For stable isotope labeling, tachyzoites of Pru, PruΔ*cdpk2* and PruΔ*pp2a-c* strains were grown in R0K0 (light), R6K4 (medium) or R10K8 (heavy) SILAC medium containing 1% dialyzed FCS for at least 8 generations to ensure efficient label incorporation, as previously described[46,47]. This approach achieved demonstrated >95% incorporation of R6, K4, R10, or K8 labels. The labeled tachyzoites of each strain were added to HFF monolayers in 175 cm² tissue culture flasks and allowed to invade cell monolayers in SILAC labeling medium for 4 h before extensive washing and addition of alkaline SILAC labeling medium (pH 8.2). Parasites were cultured in a 37 °C incubator without $CO_2$ for 4 days and the medium was replaced every day to maintain a high alkaline condition. Parasites were purified and harvested as described above, and samples were prepared for each strain from four independent experiments.

## Cell lysis and protein digestion

Parasite lysate was sonicated three minutes on ice using a high intensity ultrasonic processor (Scientz) in lysis buffer (1% SDS, 1% protease inhibitor cocktail, 1% phosphatase inhibitor for phosphorylation). The debris was removed by centrifugation at $12,000 \times g$ for 10 min at 4 °C. The supernatant was collected and protein concentration was determined with BCA kit according to the manufacturer's instructions.

For trypsin digestion, 1 volume of pre-cooled acetone was added to the protein sample. The samples were then vortexed, followed by the addition of 4 volumes of pre-cooled acetone and precipitation at −20 °C for 2 h. The protein sample was then redissolved in 200 mM TEAB and ultrasonically dispersed. Trypsin was added at 1:50 trypsin-to-protein mass ratio for the first digestion overnight. The sample was reduced with 5 mM dithiothreitol for 60 min at 37 °C and alkylated with 11 mM iodoacetamide for 45 min at room temperature in the dark. Finally, the peptides were desalted by using a Strata X SPE column.

## Phosphopeptide enrichment

Peptide mixtures were first incubated with IMAC microspheres' suspension with vibration in loading buffer (50% acetonitrile/0.5% acetic acid). To remove non-specifically adsorbed peptides, IMAC microspheres were sequentially washed with 50% acetonitrile/0.5% acetic acid and 30% acetonitrile/0.1% trifluoroacetic acid. Then, an elution buffer containing 10% NHOH was added and the enriched phosphopeptides were eluted with vibration. The supernatant containing phosphopeptides was collected and lyophilized for LC-MS/MS analysis.

## LC-MS/MS analysis and data processing

The tryptic peptides were dissolved in solvent A (0.1% formic acid and 2% acetonitrile), loaded onto a home-made reversed-phase analytical column (15-cm length, 75 μm i.d.). The gradient involved an increase from 7% to 11% solvent B (0.1% formic acid in 90% acetonitrile) over 4 min, 11% to 32% in 50 min and climbing to 80% in 3 min then holding at 80% for the last 3 min, all at a constant flow rate of 500 nL/min on an EASY-nLC 1200 system. The peptides were subjected to NSI source followed by tandem mass spectrometry (MS/MS) in Exploris 480 (Thermo) coupled online to the EASY-nLC. The electrospray voltage applied was 2.3 kV. The $m/z$ scan range was 400–1200 for full scan, and intact peptides were detected in the Orbitrap at a resolution of 60,000. Peptides were then selected for MS/MS using NCE setting as 35 and the fragments were detected in the Orbitrap at a resolution of 15,000. A data-dependent procedure alternating between MS scan was driven by 1 s cycle time method with 30 s dynamic exclusion. Automatic gain control (AGC) was set at 5E4. Fixed first mass was set as 110 $m/z$.

## Statistics and reproducibility

All statistical analyses were performed using Prism software version 7.01 (GraphPad Software Inc., CA, USA). Tables and Pie charts were visualized with Microsoft Excel. Information including the number of biological replicates, number of observations made, error bars defined and exact statistical tests used can be found in the relevant figure legends. $P$-values <0.05 were considered significant. All microscopy images are representatives of at least two independent experiments and all experiments resulted in comparable results.

## Reporting summary

Further information on research design is available in the Nature Portfolio Reporting Summary linked to this article.

## Data availability

The RNA-Seq data reported in this study are available in the short read archive (SRA) of NCBI under the accession number PRJNA819218. The phosphoproteomics data are available via ProteomeXchange with the identifier PXD033120. *Toxoplasma gondii* genome information can be found in ToxoDB (https://toxodb.org) and Eukaryotic pathogen, Vector & Host Informatics Resources can be found in VEupathDB (https://veupathdb.org). PDB file generated in the previous study and used here include 2IAE (https://www.rcsb.org/structure/2IAE). Source data are provided with this paper.

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

## Acknowledgements

We thank Prof Guo-Hua Liu from the College of Veterinary Medicine, Hunan Agricultural University for assistance with phylogenetic analysis. We also thank Dr Hu Dong from Lanzhou Veterinary Research Institute, Chinese Academy of Agricultural Sciences for assistance with 3D structure analysis. Jingjie PTM Biolab Co. Ltd (Hangzhou, China) is thanked for technical assistance with proteomics studies. The authors acknowledge the technical assistance of BGI-Shenzhen in RNA-Seq analysis. This work was supported by the National Key Research and Development Program of China (Grant Nos. 2021YFC2300800 and 2021YFC2300802), the National Natural Science Foundation of China (Grant Nos. 32002310 and 32172887) to T.T.L. and X.Q.Z., the Research Funding from Lanzhou Veterinary Research Institute (Grant No. CAA-SASTIP-JBGS-20210801) to J.L.W., the State Key Laboratory of Veterinary Etiological Biology (Grant No. SKLVEB2020YQRC01) to J.L.W., and the Fund for Shanxi "1331 Project" (Grant No. 20211331-13) to X.Q.Z. The funders had no role in the study design, data analysis, data interpretation, and the writing of this report. All authors had full access to the data in the study and accept the responsibility to submit it for publication.

## Author contributions

J.L.W., X.Q.Z., and H.M.E. conceived and designed the study. J.L.W., T.T.L., and Q.L.L. performed experiments and analyzed data with contributions from Z.W.Z. and M.W. J.L.W. and T.T.L. wrote the manuscript and produced the figures. L.D.S., H.M.E., J.L.W., and X.Q.Z. critically revised and edited the manuscript. X.Q.Z. and J.L.W. secured the funds. X.Q.Z. supervised the project. All authors reviewed and approved the final version of the manuscript.

## Competing interests

The authors declare no competing interests.
