## [Peer Review File · Nature Communications]

Reviewer comments, first round -

Reviewer #1 (Remarks to the Author):

Wang et al., report on the role of the trimeric protein phosphatase 2A holoenzyme in affecting amylopectin metabolism and stage differentiation of *T. gondii*. The authors use immunoprecipitation to show that all three necessary subunits interact. Knock-out of all three units yield similar phenotypes that are marked by starch accumulation and attenuated stage conversion. The latter is shown by quantifying the RNA expression of multiple markers by RNAseq and DBA staining. PP2A-null parasites are avirulent in mice in high doses and do not form tissue cysts in the brain. PP2A acts as a phosphatase and mutagenesis of conserved motifs aborts this function. The authors identify differences in the phosphoproteome between PP2A-null, wild-type and CDPK2-null strains and investigate the role of these proteins by reverse genetics.

The paper reports an impressive amount of data that is generally sound and supports the overall conclusions. In my view this paper represents a commendable attempt to approach the challenging task of dissecting a highly multi-factorial function of the PP2A in *T. gondii* and will be of great interest to the wider community. I summarize the main issues that can be addressed in writing or straight forward experiments below.

(1) The link between PP2A protein function and the observed phenotypes is not very strong. The authors investigate the function of PP2A via reverse genetic mutagenesis of conserved motifs and show that PP2A-c-null mutants harbour a hyperphosphorylated proteins. Inspecting the data it appears that hyper- and hypo-phosphorylated proteins are roughly in balance in the PP2A-c ko indicating many secondary effects and not an overall hyperphosphorylation, which may be indicative of the PP2A phosphatase function.

The authors explain the PP2A-ko phenotype, which is certainly a result of the functional regulation of many proteins, via genetic data on the role of CDPK2 phosphosites on amylopectin metabolism and via the impact of four hyperphosphorylated proteins on differentiation. While CDPK2 is already known to regulate amylopectin pools in *T. gondii*, the identification of functionally relevant phosphosites is valuable and also convincingly shown. However, there is no attempt to quantify the extent in which *cdpk2* dysregulation is responsible for the amylopectin accumulation on PP2A-null mutants.

In addition, while implication of the four hyperphosphorylated proteins in bradyzoite differentiation is new, their ablation is not well documented. Controls such as solvent controls (for mAID dependent approaches), western blots and complementation experiments are missing. One of these four proteins is BFD2 (<https://www.biorxiv.org/content/10.1101/2022.04.06.487076v1>) and the preprint may be cited. Also, an attempt to quantify their contribution to the PP2A-ko phenotype is missing.

The authors should close this gap between PP2A protein function and phenotype by providing data that directly demonstrates the phosphatase activity of PP2A and/or links hyperphosphorylation to amylopectin/differentiation.

(2) The conclusion that PP2A is required for tissue cyst formation *in vivo* may be over reaching as the absence of virulence may prevent parasites from disseminating to the brain. However, represents a challenging problem which the authors already tried to address by using multiple doses of parasites and two mouse strains. In absence of more definite data linking clinical signs of infection with the absence of cysts the authors may consider toning down this conclusion.

Minor issues

Line 54: consider adding spontaneous differentiation in certain cell types such as C2C12, KD3

Fig 1B: Difficult to discern individual bradyzoites in images. Bradyzoites are very young with three days

Fig 1: Western Blots or PCRs showing correct integration of tags missing

Fig 2D: y-axis designation is missing. What has the "relative plaque size" been normalized to?

Fig S3C the PP2a ko parasites appear to form large vacuoles that may be held together by a non-DBA positive cyst wall as shown here (<https://journals.plos.org/plospathogens/article?id=10.1371/journal.ppat.1003331>). Can the authors confirm the absence of a cyst wall structure by EM or other methods?
Line 212, 217: References to Fig 4A and Fig 4B in this sections should point to Fig 5A,B
Line 224: Fig 5 references should point to Fig 4.
Fig S4E: it remains unclear in which mice the cyst count was performed.

Fig S6A should show the slower migration of phosphorylated CDPK2 protein. This difference appears very small at best and not easily recognizable. Can the authors show another blot instead? There also appear other bands to be cropped.
Fig 6B: The authors do not explain the rationale for creating PP2A and GP double mutants and the relevance of this experiment might be hard to grasp for the uninitiated reader.

Please add images of PCRs verifying *T. gondii* mutants.

Reviewer #2 (Remarks to the Author):

The work presented here contributes to our understanding of the regulatory factors of *T. gondii* that contribute to the accumulation of cytoplasmic starch granules that are a hallmark of the dormant bradyzoite stage. In particular, the authors shed new light on how the holoenzyme PP2A contributes to the phosphorylation balance that controls multiple protein functions. Using quantitative phosphoproteomics, the authors demonstrated that dephosphorylation of calcium-dependent protein kinase 2 at S679 contributes to the regulation of amylopectin metabolism. CDPK2 has previously been shown to regulate starch storage and is essential for the development of viable cysts in *Toxoplasma gondii* (Uboldi, A.D. et al. *Cell Host Microbe*, 2015). In this context, this study provides new insights into the regulation of amylopectin. They also identified several novel and promising proteins under the control of PP2A, including BFD2 (TGME49_311100). Recently, parasites lacking BFD2 were shown to be unable to induce BFD1, a master regulator controlling chronic differentiation (Licon MH et al. bioRxiv preprint). The manuscript provides a fairly comprehensive analysis of the PP2A phosphoproteome and the likely target proteins that play critical roles in parasite differentiation. However, a few important points dampen enthusiasm.

Licon MH et al. bioRxiv preprint : <https://doi.org/10.1101/2022.04.06.487076>

Major comments:

- The authors apparently succeeded in constructing KO of PP2A-A (TGME49_315670) and PP2A-B (TGME49_246510), although these genes have very low fitness scores of -4.03 and -2.32, respectively (Sidik SM, 2016), indicating their essentiality. I am very skeptical about the conclusions that can be drawn from this. Conceptually, only PP2A-C (TGME49_224220) could be interrupted since it has a mild fitness score of -1.5. The essentiality is evident in their plaque assays that show fewer and smaller plaques in the three delta-pp2a mutants compared to their parental Fig. 2C. It is difficult to draw conclusions from strains that are so unlikely to complete the lytic cycle and whose phenotype is difficult to interpret. Why not use the inducible mAID system as the authors did for the other essential genes (Fig. 7) ?
- The authors overlooked or ignored another gene, TGME49_286210, which encodes a protein structurally similar to PP2A-C (PDB: 2ie3_C) and has a fitness score of -4.79, indicating stronger and expected essentiality. Is it possible that this protein is part of another PP2A holoenzyme?
- The immunoprecipitation experiments (Fig. S1) lack a comparison between input and flow-through to evaluate the left over after IP. Moreover, a robust enrichment, if any, should compare the eluates to 10% of the input deposited in the gel. The best way to prove that the three subunits form a complex core would have been to demonstrate their co-elution on a size exclusion column.
- Since KO parasites have a strong growth defect in vitro, what is the relevance of studying their phenotype in mice, as we know that the differences in type II load can cause large differences in the outcome of infection. It is known that the Pruku80 strain forms few cysts, as shown by the

authors when mice were infected with the parental strain. Thus, the claim that the PP2A holoenzyme is essential for bradyzoite differentiation must be supported by using a genetic background more conducive to cyst formation (ME49, 76K strains...).

Minor comments:

- No legend was provided for supplemental data S1, which complicates the analysis of the mass spectrometry data. Is abundance or iBAQ given in data S1? How was the ranking generated?
- In lines 103-107, the authors write "To determine which regulatory subunit binds to the core enzyme, immunoprecipitation (IP) was performed. First, a 6×Myc epitope tag was fused to the C-terminus of PP2A-B and PP2A-A Pru strains that were then induced to form bradyzoites over four days in alkaline condition. The PP2A subunits were immunoprecipitated with anti-Myc-conjugated magnetic beads and then analyzed using LC-MS." One may wonder why the partnership between the different subunits was not determined at the tachyzoite stage, where the proteins are expressed.
- The subcellular co-localization experiments are not resolving enough to say that the proteins occur together in the parasite. At least confocal reconstructions or even super-resolution STED would be required to make such a statement.

Re: Manuscript NCOMMS-22-12391.R1

Responses to comments and suggestions of Reviewer #1

General comments:

Wang et al., report on the role of the trimeric protein phosphatase 2A holoenzyme in affecting amylopectin metabolism and stage differentiation of *T. gondii*. The authors use immunoprecipitation to show that all three necessary subunits interact. Knock-out of all three units yield similar phenotypes that are marked by starch accumulation and attenuated stage conversion. The latter is shown by quantifying the RNA expression of multiple markers by RNAseq and DBA staining. PP2A-null parasites are avirulent in mice in high doses and do form tissue cysts in the brain. PP2A acts as a phosphatase and mutagenesis of conserved motifs aborts this function. The authors identify differences in the phosphoproteome between PP2A-null, wild-type and CDPK2-null strains and investigate the role of these proteins by reverse genetics.

The paper reports an impressive amount of data that is generally sound and support the overall conclusions. In my view this paper represents a commendable attempt to approach the challenging task of dissecting a highly multi-factorial function of the PP2A in *T. gondii* and

will be of great interest to the wider community. I summarize the main issues that can be addressed in writing or straight forward experiments below.

Response: We thank the reviewer very much for the favorable comments and constructive suggestions on our MS. We have revised the MS strictly according to these constructive comments and suggestions.

Comment: The link between PP2A protein function and the observed phenotypes is not very strong. The authors investigate the function of PP2A via reverse genetic mutagenesis of conserved motifs and show that PP2A-c-null mutants harbour a hyperphosphorylated proteins. Inspecting the data, it appears that hyper- and hypo-phosphorylated proteins are roughly in balance in the PP2A-c ko indicating many secondary effects and not an overall hyperphosphorylation, which may be indicative of the PP2A phosphatase function.

Response: We thank the reviewer very much for the constructive suggestions. Indeed, there were 80 phosphosites on 58 proteins that were significantly hypo-phosphorylated upon PP2A-C depletion (Supplementary Data 4). It is likely to be secondary affects caused possibly by PP2A-c activating other phosphatases or suppressing kinases. We have added this point to the revised MS.

Comment: The authors explain the PP2A-ko phenotype, which is certainly a result of the functional regulation of many proteins, via genetic data on the role of CDPK2 phosphosites on amylopectin metabolism and via the impact of four hyperphosphorylated proteins on differentiation. While CDPK2 is already known to regulate amylopectin pools in *T. gondii*, the identification of functionally relevant phosphosites is valuable and also convincingly shown. However, there is no attempt to quantify the extent in which *cdpk2* dysregulation is responsible for the amylopectin accumulation on PP2A-null mutants.

Response: We thank the reviewer very much for the constructive suggestions. To address this interesting point, we have quantified the extent to which CDPK2 dysregulation is responsible for amylopectin accumulation in PP2A-null mutants. The new results showed that with the complementation of phosphomimetic CDPK2^{S679E} and phosphoablative CDPK2^{S679A} in *PruΔpp2a-cΔcdpk2* null mutants, there was an accumulation of large amylopectin granules in both mutants, suggesting that a balance between phosphorylation and dephosphorylation of S679 is required to maintain CDPK2 activity. We have added this information to the revised MS.

Comment: In addition, while implication of the four hyperphosphorylated proteins in bradyzoite differentiation is new, their ablation is not well documented. Controls such as solvent controls (for mAID dependent approaches), Western blots and complementation experiments are missing. One of these four proteins is BFD2 (<https://www.biorxiv.org/content/10.1101/2022.04.06.487076v1>) and the preprint may be cited. Also, an attempt to quantify their contribution to the PP2A-ko phenotype is missing.

The authors should close this gap between PP2A protein function and phenotype by providing data that directly demonstrates the phosphatase activity of PP2A and/or links hyperphosphorylation to amylopectin/differentiation.

Response: We thank the reviewer very much for the constructive suggestions. We have added new data on the solvent controls (for mAID dependent approaches), Western blots and complementations. We are unable to complement the Pru Δ 298610 strain due to the large CDS of the TGME49_298610. However, we used the inducible mAID system to knockout TGME49_298610 protein and similar phenotypes were observed. We have added the citation for the suggested preprint of BFD2 as suggested.

The complementation of phosphomimetic CDPK2S679E and phosphoablative CDPK2S679A in the PP2A-c null mutants did not alter the starch accumulation caused by disruption of PP2A. Therefore, we inferred that the four hyperphosphorylated proteins may have the same effect on the PP2A-c null mutants. In addition, some transcription factors involved in parasite differentiation were only expressed in the bradyzoite stage such as the BFD1, and their phosphorylation level is difficult to assess in Pru Δ pp2a-c strain because Pru Δ pp2a-c cannot form bradyzoites. Therefore, there may be other proteins involved in parasite differentiation and further studies are still needed to elucidate the mechanisms that control PP2A-mediated parasite differentiation.

The balance between phosphorylation catalyzed by kinases and de-phosphorylation mediated by phosphatases is needed to maintain the protein function. In this study, one of the key proteins involved in de-phosphorylation was identified as PP2A; however, the respective kinases are unknown. Hence, it was difficult to provide data that directly demonstrates the phosphatase activity of PP2A to these proteins. Further studies are required to identify the kinases and evaluate their relationship.

Comment: The conclusion that PP2A is required for tissue cyst formation *in vivo* may be overreaching as the absence of virulence may prevent parasites from disseminating to the brain. However, represents a challenging problem which the authors already tried to address by using multiple doses of parasites and two mouse strains. In absence of more definite data linking clinical signs of infection with the absence of cysts the authors may consider toning down this conclusion.

Response: We thank the reviewer very much for the constructive suggestion. We also knocked out the PP2A-C in the ME49 strain which is more conducive to cyst formation, and our results showed that ME49 Δ pp2a-c also failed to form brain cysts in mice, although ME49 Δ pp2a-c can reach to the brain as detected by PCR.

Minor issues

Line 54: consider adding spontaneous differentiation in certain cell types such as C2C12, KD3.

Response: We thank the reviewer very much for the constructive suggestions. We have added

the spontaneous differentiation in C2C12 cells as suggested. We found that *PruΔpp2a-c* failed to form cysts in C2C12 cells.

Fig 1B: Difficult to discern individual bradyzoites in images. Bradyzoites are very young with three days.

Response: We thank the reviewer very much for the constructive suggestion. We have presented 7-day-old cysts and the parasites were detected by IMC1 staining.

Fig 1: Western Blots or PCRs showing correct integration of tags missing.

Response: We have added the missing information.

Fig 2D: y-axis designation is missing. What has the “relative plaque size” been normalized to?

Response: Apologies for this oversight. The y-axis designation was overshadowed by Figure 2C, but we have adjusted this appropriately.

Fig S3C the PP2a ko parasites appear to form large vacuoles that may be held together by a non-DBA positive cyst wall as shown here (<https://journals.plos.org/plospathogens/article?id=10.1371/journal.ppat.1003331>). Can the authors confirm the absence of a cyst wall structure by EM or other methods?

Response: The absence of a cyst wall structure was shown by EM. In addition, the inability of *Δpp2a* parasites to form cysts was confirmed by BAG1 and s-WGA (succinylated wheat germ agglutinin) staining.

Line 212, 217: References to Fig 4A and Fig 4B in these sections should point to Fig 5A,B.

Response: This was revised accordingly.

Line 224: Fig 5 references should point to Fig 4.

Response: This was revised accordingly.

Fig S4E: it remains unclear in which mice the cyst count was performed.

Response: This was revised accordingly.

Fig S6A should show the slower migration of phosphorylated CDPK2 protein. This difference appears very small at best and not easily recognizable. Can the authors show another blot instead? There also appear other bands to be cropped.

Response: This was revised accordingly.

Fig 6B: The authors do not explain the rationale for creating PP2A and GP double mutants and the relevance of this experiment might be hard to grasp for the uninitiated reader.

Response: We have revised this section to clarify this point in the revised MS.

Please add images of PCRs verifying *T. gondii* mutants.

Response: Images were added as requested.

Responses to comments and suggestions of Reviewer #2

General comments:

The work presented here contributes to our understanding of the regulatory factors of *T. gondii* that contribute to the accumulation of cytoplasmic starch granules that are a hallmark of the dormant bradyzoite stage. In particular, the authors shed new light on how the holoenzyme PP2A contributes to the phosphorylation balance that controls multiple protein functions. Using quantitative phosphoproteomics, the authors demonstrated that dephosphorylation of calcium-dependent protein kinase 2 at S679 contributes to the regulation of amylopectin metabolism. CDPK2 has previously been shown to regulate starch storage and is essential for the development of viable cysts in *Toxoplasma gondii* (Uboldi, A.D. et. al. Cell Host Microbe, 2015). In this context, this study provides new insight into the regulation of amylopectin. They also identified several novel and promising proteins under the control of PP2A, including BFD2 (TGME49_311100). Recently, parasites lacking BFD2 were shown to be unable to induce BFD1, a master regulator controlling chronic differentiation (Licon MH et al. bioRxiv preprint). The manuscript provides a fairly comprehensive analysis of the PP2A phosphoproteome and the likely target proteins that play critical roles in parasite differentiation. However, a few important points dampen enthusiasm. Licon MH et al. bioRxiv preprint: <https://doi.org/10.1101/2022.04.06.487076>

Response: We thank the reviewer very much for the favorable comments and constructive suggestions on our MS. We have revised the MS strictly according to these comments and suggestions.

Major comments:

- The authors apparently succeeded in constructing KO of PP2A-A (TGME49_315670) and PP2A-B (TGME49_246510), although these genes have very low fitness scores of -4.03 and -2.32, respectively (Sidik SM, 2016), indicating their essentiality. I am very skeptical about the conclusions that can be drawn from this. Conceptually, only PP2A-C (TGME49_224220) could be interrupted since it has a mild fitness score of -1.5. The essentiality is evident in their plaque assays that show fewer and smaller plaques in the three delta-pp2a mutants compared to their parental Fig. 2C. It is difficult to draw conclusions from strains that are so unlikely to complete the lytic cycle and whose phenotype is difficult to interpret. Why not use the inducible mAID system as the authors did for the other essential genes (Fig. 7)?

Response: We thank the reviewer very much for the constructive suggestions on our manuscript. We have also utilized the auxin-inducible degron (AID) system for conditional

deletion of the PP2A-C, PP2A-B and PP2A-A. Similar results were observed in the experiments conducted with $\Delta pp2a-c$, $\Delta pp2a-b$ and $\Delta pp2a-a$ strains.

- The authors overlooked or ignored another gene, TGME49_286210, which encodes a protein structurally similar to PP2A-C (PDB: 2ie3_C) and has a fitness score of -4.79, indicating stronger and expected essentiality. Is it possible that this protein is part of another PP2A holoenzyme?

Response: We utilized the auxin-inducible degron (AID) system for conditional knockout of the TGME49_286210 and the results showed that TGME49_286210 is essential for parasite replication, but not for bradyzoite differentiation or starch metabolism, suggesting that this protein is less likely to be a part of the PP2A holoenzyme.

- The immunoprecipitation experiments (Fig. S1) lack a comparison between input and flow-through to evaluate the left over after IP. Moreover, a robust enrichment, if any, should compare the eluates to 10% of the input deposited in the gel. The best way to prove that the three subunits form a complex core would have been to demonstrate their co-elution on a size exclusion column.

Response: We have performed this experiment again and compared the eluates to 50% of the input deposited in the gel as suggested. We have revised the MS accordingly.

- Since KO parasites have a strong growth defect in vitro, what is the relevance of studying their phenotype in mice, as we know that the differences in type II load can cause large differences in the outcome of infection. It is known that the Pruku80 strain forms few cysts, as shown by the authors when mice were infected with the parental strain. Thus, the claim that the PP2A holoenzyme is essential for bradyzoite differentiation must be supported by using a genetic background more conducive to cyst formation (ME49, 76K strains...).

Response: We thank the reviewer very much for the constructive suggestions. We have also knocked out the PP2A-C in the ME49 strain, and our results showed that ME49 $\Delta pp2a-c$ also failed to form brain cysts in mice.

Minor comments:

- No legend was provided for supplemental data S1, which complicates the analysis of the mass spectrometry data. Is abundance or iBAQ given in data S1? How was the ranking generated?

Response: We have added the abundance and other information of the mass spectrometry in the supplemental data S1. We have also added the legend of this and other supplementary data files.

- In lines 103-107, the authors write “To determine which regulatory subunit binds to the core enzyme, immunoprecipitation (IP) was performed. First, a 6×Myc epitope tag was fused to

the C-terminus of PP2A-B and PP2A-A Pru strains that were then induced to form bradyzoites over four days in alkaline condition. The PP2A subunits were immunoprecipitated with anti-Myc-conjugated magnetic beads and then analyzed using LC-MS.” One may wonder why the partnership between the different subunits was not determined at the tachyzoite stage, where the proteins are expressed.

Response: In this study, we have investigated the partnership between the different subunits in both stages. We have used Co-IP for the tachyzoites and IP-MS for the bradyzoites to detect the partnership between the different subunits.

- The subcellular co-localization experiments are not resolving enough to say that the proteins occur together in the parasite. At least confocal reconstructions or even super-resolution STED would be required to make such a statement.

Response: The co-localization experiments were performed by confocal and new figures have been added.

We have carefully examined the entire MS and corrected other minor mistakes and typo-errors.

Once again, thank you and thanks to the reviewers for the favorable comments and constructive recommendations that all authors have greatly appreciated.

We hope that the MS has been revised to your satisfaction. We are looking forward to receiving your editorial decision soon.

Sincerely yours,

Xing
Xing-Quan Zhu, BVSc, MVSc, PhD
Professor & Head, Laboratory of Parasitic Diseases,
College of Veterinary Medicine,
Shanxi Agricultural University,
Taigu, Shanxi Province 030801,
People’s Republic of China
Email: xingquanzhu1@hotmail.com

Reviewer comments, second round -

Reviewer #1 (Remarks to the Author):

The revised Wang et al manuscript incorporates a substantial amount of new data including several genetic complementations and CDPK2-phosphomimetic mutants in a PP2A-Cko background. These data solidify conclusions and are valuable to further define the roles of both proteins in starch metabolism and suggest involvement of other kinases as well. These data along with many clarifications in writing result in a much improved manuscript that in my view can be accepted for publication.

Reviewer #2 (Remarks to the Author):

This study is a thorough analysis of the protein subunits of the phosphatase 2A holoenzyme. They find that protein phosphatase A contributes to the regulation of amylopectin metabolism and bradyzoite development. All the experiments are carefully performed with good controls and beautifully presented as figures. The authors were thoroughly responsive to all of the suggestions from the previous critiques. In two places, qPCR of the B1 brain found infected brains, and complementation with the 3HA-tag at the C-terminus, the authors state "data not shown". These are important controls, so it would be best to include these data in the supplemental material.

Responses to reviewers' comments

Reviewer #1 (Remarks to the Author):

The revised Wang et al manuscript incorporates a substantial amount of new data including several genetic complementations and CDPK2-phosphomimetic mutants in a PP2A-Cko background. These data solidify conclusions and are valuable to further define the roles of both proteins in starch metabolism and suggest involvement of other kinases as well. These data along with many clarifications in writing result in a much improved manuscript that in my view can be accepted for publication.

Response: We thank the reviewer very much for the favorable comments on our revised manuscript and for recommending the acceptance of our manuscript for publication.

Reviewer #2 (Remarks to the Author):

This study is a thorough analysis of the protein subunits of the phosphatase 2A holoenzyme. They find that protein phosphatase A contributes to the regulation of amylopectin metabolism and bradyzoite development. All the experiments are carefully performed with good controls and beautifully presented as figures. The authors were thoroughly responsive to all of the suggestions from the previous critiques. In two places, qPCR of the B1 brain found infected brains, and complementation with the 3HA-tag at the C-terminus, the authors state “data not shown”. These are important controls, so it would be best to include these data in the supplemental material.

Response: We thank the reviewer very much for the favorable comments and the constructive suggestions on our manuscript. We have added the data on the PCR of *BI* gene in the infected brains in **Supplementary Fig. 5i** and the complementation with the 3HA-tag at the C-terminus in **Supplementary Fig. 8e**, respectively, as recommended.